# Extension of AVHRR-based climate data records: Exploring ways to simulate AVHRR radiances from Suomi-NPP VIIRS data

Karl-Göran Karlsson, Nina Håkansson, Salomon Eliasson, Erwin Wolters, Ronald Scheirer

Research and Development Department, Swedish Meteorological and Hydrological Institute (SMHI), Folkborgsvägen 17, 60176 Norrköping, Sweden

*Correspondence to*: Karl-Göran Karlsson (karl-goran.karlsson@smhi.se)

**Abstract.**

The long series of multispectral measurements from the Advanced Very High Resolution Radiometer (AVHRR), which began
in 1979, is now approaching its end, with the last remaining AVHRR sensor currently operating aboard EUMETSAT's Metop-C satellite. Several Climate Data Records (CDRs) built on AVHRR data now face the end of their observational record. However, since many modern imagers contain AVHRR-heritage spectral channels, a potential for extension of these AVHRR-based climate data records exists. This study investigates the possibility to simulate original National Oceanic and Atmospheric Administration (NOAA)-19 AVHRR channels from the Suomi National Polar-orbiting Platform (NPP) Visible Infrared
Imaging Radiometer Suite (VIIRS) radiances using collocated AVHRR/VIIRS datasets from 2012-2013. Spectral Band Adjustments (SBAs) were derived using linear regression and neural networks (NNs). The NN approach produced the best results, and separating daytime from night-time conditions when simulating AVHRR channel 3B at 3.7 µm was key. Furthermore, daytime radiance corrections in this channel must depend on actual surface and cloud reflectances to be realistic, which was only achieved by the NN approach.

The cloud mask, cloud top height, and cloud phase products were produced from the simulated AVHRR radiances using the same retrieval methods for NOAA-19 data used to compile the CLARA-A3 CDR. CLARA-A3 is the third edition of the EUMETSAT Climate Monitoring Satellite Application Facility (CM SAF) CDR with cloud parameters, surface albedo, surface radiation, and Top of Atmosphere (TOA) radiation products from AVHRR. Products were validated using Cloud-Aerosol Lidar and Infrared Pathfinder Satellite Observations - Cloud-Aerosol Lidar with Orthogonal Polarization (CALIPSO-
CALIOP) cloud products and agreed well with original CLARA-A3 products, with the best results provided by the NN simulation approach. The NN-based approach best reproduced the corresponding products for cloud optical thickness (COT), cloud effective radius (CRE), liquid water path (LWP), and ice water path (IWP).

The CLARA-A3 CDR will be complemented and extended with VIIRS-based products to cover the period 1979-2024 (46 years). This edition will be known as CLARA-A3.5. Future extensions and editions can follow a similar approach by applying
the same radiance simulation method to collocated data from the Metop-C AVHRR and the Metop-Second Generation (SG) METimage sensors, the first satellite of the latter scheduled for launch in August 2025. Successful simulation of AVHRR radiances from METimage and VIIRS data enables the CLARA CDR extension for several decades.

# 1 Introduction

Successful climate monitoring depends on the availability of long observational time series from reliable and stable observation platforms and sensors. Observations with very long temporal coverage (i.e., on century scales) have been mainly restricted to measurements from land-based surface stations and mostly limited to 2-meter temperature measurements (e.g., Morice et al., 2021). For even longer perspectives, various proxy observations must be used (e.g., tree ring and sediment climatologies; Anchukaitis et al., 2017).

However, to fundamentally describe and understand climate and climate change, global observations at high spatio-temporal resolution are needed. Furthermore, a full range of different meteorological parameters need to be covered. The first steps towards realizing an observation system with truly global coverage were taken when information from polar and geostationary satellites was introduced in the 1960s. These sensors were later upgraded and introduced in operational missions by the end of the 1970s (Kidd et al., 2009; Giri et al., 2025). Additional observations with better coverage of ocean surfaces and upper air were introduced through various technological developments (Lin and Yang, 2020; WMO, 2024; NDACC, 2024). Furthermore, the systematic use of radiation network measurement data (NDACC, 2021) from active and passive remote sensing instruments at surface stations and on space platforms is now standard (Thies and Bendix, 2011; eoPortal, 2024). All these developments made it possible to compile comprehensive and consistent climate datasets by synthesizing data from all types of observation platforms in reanalysis datasets (Hersbach et al., 2020).

Reanalysis datasets are undoubtedly capable of providing the best possible description of the Earth's atmospheric and surface state evolution, at least over the last 3-5 decades, with access to a multitude of global observations and the use of a physically consistent methodology based on model physics constraints. However, because of the use of data from an ever-changing observation system, not least after the introduction of several new or improved satellite sensors over the last few decades, the uncertainty regarding the existence and magnitude of climate trends in reanalysis results is still considerable (Bengtsson et al., 2004; Thorne et al., 2005; de Padua and Ahn, 2024; Tarek et al., 2021). In addition, some parameters of great importance for the Earth's radiation balance are not yet fully assimilated from observations. This concern, in particular, cloudiness and cloud properties (Yao et al., 2020). Furthermore, the reanalysis dependency on physical constraints from the current Numerical Weather Prediction (NWP) model means that the results are not completely independent, since model physics cannot be considered as perfectly describing the real atmosphere/Earth system (as pointed out by Roebeling et al., 2025).

With this background, the value of a long time-series of single sensor observations or measurements for climate studies would still be high. This concern, not least, Climate Data Records (CDRs) from satellite platforms, where several of them now cover considerably longer periods than the standard World Meteorological Organisation (WMO) climatological 30-year period. The Advanced Very High Resolution Radiometer (AVHRR), operating onboard polar satellites since 1978, provides the longest available time series of observations from meteorological satellite imagery. The third edition of the European Organization for the Exploitation of Meteorological Satellites (EUMETSAT) Climate Monitoring Satellite Application Facility (CM SAF) CDR with cloud parameters, surface albedo, surface radiation, and Top of Atmosphere (TOA) radiation products from AVHRR

(CLARA-A3, see Karlsson et al., 2023) covers more than four decades of AVHRR observations. Neither AVHRR radiances nor AVHRR-derived cloud and radiation products have yet been assimilated in reanalysis datasets, which is an additional argument for their value as an independent observation dataset. However, the last AVHRR instrument was launched with the EUMETSAT satellite Metop-C in 2018. Thus, the AVHRR era will soon be over.

This paper investigates methods to extend the CLARA CDR with data from the AVHRR successor, the Visible Infrared Imaging Radiometer Suite (VIIRS), which is now operational on current polar meteorological satellites from NOAA. If methods are successful, the CLARA CDR can be extended by at least 2-3 decades. The paper studies two approaches to simulate AVHRR radiances from VIIRS, using Spectral Band Adjustment Factors (SBAFs): 1. using linear regression 2. a method using a MultiLayer Perceptron (MLP) neural network. As a further test of success besides ordinary radiance-to-

radiance comparisons, the simulated radiances are used to produce the CLARA cloud properties, which are then validated using independent cloud observations from the Cloud-Aerosol Lidar with Orthogonal Polarisation (CALIOP) onboard the Cloud-Aerosol Lidar and Infrared Pathfinder Satellite Observation (CALIPSO) satellite (Winker et al., 2009).

Section 2 describes the methodological background and the used datasets. The methodology is then described in detail in Sect. 3, followed by results in Sect. 4. Further analysis and discussions are presented in Sect. 5, with conclusions given in Sect. 6.

**2 Methodological background and selected datasets**

**2.1 Introduction to Spectral Band Adjustment Methods**

Adjusting measurements after introducing a slightly modified or new sensor version poses a longstanding challenge that has received considerable attention over the years. Most well-known are the activities of the Global Space-based Inter-Calibration System (GSICS, WMO 2025), where the primary goal is to ensure a homogeneous behaviour of measurement time series from

a particular sensor or spectral channel. We call this adjustment "Inter-calibration" (Chander et al., 2013a) and the purpose here is to provide a homogenous data record without artificial discontinuities. The Fidelity and uncertainty in climate data records from Earth Observations project (FIDUCEO) emphasized the difference between homogenized and harmonized datasets (Giering et al., 2019) with relevance for the CDR compilation. Harmonized data would imply corrections to a measurement based on high-quality reference measurements, thus providing the best possible estimation of the measured radiance. This

correction would still allow differences to a similar instrument having slightly different spectral responses. However, for a CDR, which should allow for climate trends estimation, homogenized data seemingly should be the best choice. Homogenized data for a CDR means that measurements are corrected with respect to one particular sensor in the measurement series instead of to one high quality reference sensor. On the other hand, this could also lead to sensor accuracy violation (if the various sensors have significant differences in spectral response). Thus, there are pros and cons of both spectral adjustment

methods and any of them shall be applied with caution. An important aspect is also that radiance differences between two sensors might be caused by additional factors other than differences in spectral responses, e.g. radiance biases or calibration errors.

When focusing on the current problem to simulate AVHRR from VIIRS radiances based solely on spectral response differences, no spectral adjustment methodology will ever be able to simulate AVHRR channels perfectly, since some parts of the spectrum covered by another AVHRR-heritage sensor channel are simply not observed by the corresponding AVHRR channel (and vice versa). However, if channel differences are not very large, corrections may be sufficient, depending on the intended applications. Piontek et al. (2023) estimated that linear SBAFs can explain more than 80 % of the variance, but the efficiency depends on the selected channels.

For many years, the standard methodology to handle these spectral band adjustments has been to calculate SBAFs. These can be derived from direct inter-comparisons of spatio-temporally collocated measurements from the two sensors (for example as described by Meirink et al., 2013). Relations based on SBAFs can be either linear (Chander et al., 2013b) or non-linear and sometimes more complicated, involving more channels than the targeted spectral channel (Villaescusa-Nadal et al., 2019; Claverie, 2023). When collocations are not possible, spectrometer data can be used for calculations, most often relying on data from the SCanning Imaging Absorption spectroMeter for Atmospheric CHartographY (SCIAMACHY, Bovensmann et al., 1999) or the Infrared Atmospheric Sounding Interferometer (IASI, Blumstein et al., 2004) for meteorological applications. Hyperspectral observations are then convolved with the narrow-band Spectral Response Function (SRF) to calculate the SBAFs (Piontek et al., 2023). The NASA satellite cloud and radiation property retrieval system (SatCORPS) SBAF tool is a comprehensive and widely used web-based tool based on this technique, providing SBAFs for a wide range of sensors and satellites (NASA, 2016; Scarino et al., 2016).

## 2.2 The challenge: Bridging differences between the spectral channels of AVHRR and VIIRS

Table 1 lists the AVHRR channels simulated in this study. Notice that our reference sensor is the third version of this sensor (AVHRR/3) as carried by the NOAA-19 satellite. The choice of the AVHRR on NOAA-19 as our reference sensor is natural, since we want to replace the loss of NOAA-19 observations in the afternoon orbit after 2012 due to orbital drift. Section 2.5 provides an even stronger motivation for choosing NOAA-19. It should also be mentioned that the reference radiances for NOAA-19 AVHRR should be considered as harmonized data, since their quality and evolution over time has been optimized for this particular AVHRR sensor by a method described by Heidinger (2018).

In this study, we are not interested in simulating AVHRR channel 3A, as shown in Table 1. The reason is that satellites carrying the VIIRS sensor follow an afternoon orbit, a sun-synchronous path with a daytime equator crossing shortly after noon. For all earlier satellites used in the CLARA-A3 CDR, where AVHRR is in a similar orbit to VIIRS, only AVHRR Channel 3B was available (active). AVHRR observations have a swath width of 2600 km, and the horizontal resolution is approximately 1.1 km at nadir. However, it is much coarser (approximately 6 km) at the swath edges. Cracknell (1997) provides more details on the AVHRR imager.

**Table 1: Main AVHRR/3 sensor spectral characteristics. To be noticed is that AVHRR Channel 3A (marked in italics) is not subject to spectral conversion here (see text for explanation).**

| Channel name | Central wavelength | Spectral interval |
| --- | --- | --- |
| Channel 1 | 0.630 µm | 0.58-0.68 µm |
| Channel 2 | 0.862 µm | 0.725-1.00 µm |
| *Channel 3A* | *1.61µm* | *1.58-1.64 µm* |
| Channel 3B | 3.74 µm | 3.55-3.93 µm |
| Channel 4 | 10.80 µm | 10.3-11.3 µm |
| Channel 5 | 12.00 µm | 11.5-12.5 µm |

Table 2 gives the complete set of medium resolution channels (M-channels) of the VIIRS imager (described in more detail by Hillger et al., 2013). The swath width is 3,000 km, the horizontal resolution is 750 m at the nadir, and only slightly less (1.6 km) at the swath edges due to an oversampling scanning technique that is different from AVHRR. The AVHRR-heritage channels are marked in blue in Table 2. In theory, AVHRR channel 2 may be simulated using a combination of channels M6 and M7. However, saturation problems with channel M6 (as reported by Cao et al., 2013) produce unrealistic measurements, making it unsuitable for this purpose.

**Table 2: Main spectral characteristics of the medium resolution (M) channels of the VIIRS sensor. AVHRR-heritage channels are marked in bold. Notice that the channels marked in bold italics are not used (see text for explanation).**

| VIIRS channel name | Central wavelength | Spectral interval | Corresponding AVHRR channel |
|---|---|---|---|
| *M1* | *0.412 μm* | *0.402-0.422 μm* | - |
| *M2* | *0.445 μm* | *0.436-0.454 μm* | - |
| *M3* | *0.488 μm* | *0.478-0.498 μm* | - |
| *M4* | *0.555 μm* | *0.545-0.565 μm* | - |
| **M5** | **0.672 μm** | **0.662-0.682 μm** | **Channel 1** |
| ***M6*** | ***0.746 μm*** | ***0.739-0.754 μm*** | ***Channel 2*** |
| **M7** | **0.865 μm** | **0.846-0.885 μm** | **Channel 2** |
| *M8* | *1.240 μm* | *1.230-1.250 μm* | - |
| *M9* | *1.378 μm* | *1.371-1.386 μm* | - |
| ***M10*** | ***1.610 μm*** | ***1.580-1.640 μm*** | ***Channel 3A*** |
| *M11* | *2.250 μm* | *2.225-2.275 μm* | - |
| **M12** | **3.700 μm** | **3.691-3.709 μm** | **Channel 3B** |
| *M13* | *4.050 μm* | *3.973-4.128 μm* | - |
| *M14* | *8.550 μm* | *8.400-8.700 μm* | - |
| **M15** | **10.763 μm** | **10.263-11.263 μm** | **Channel 4** |
| **M16** | **12.013 μm** | **11.538-12.488 μm** | **Channel 5** |

For a better visualization of the differences between the two sensor's channels, we can study the differences in spectral
responses as illustrated in Fig. 1. It is clear that there are indeed substantial differences for most channels, except possibly for AVHRR channels 4 and 5. However, the different wavelength scales at the x-axes in the plots in Fig. 1 tend to exaggerate differences for some channels (e.g., AVHRR channel 1) and underrate differences for other channels (e.g., AVHRR channel 2). AVHRR channel 2 shows the most significant difference, with a broader spectral coverage than the VIIRS AVHRR heritage channels (M6 and M7).

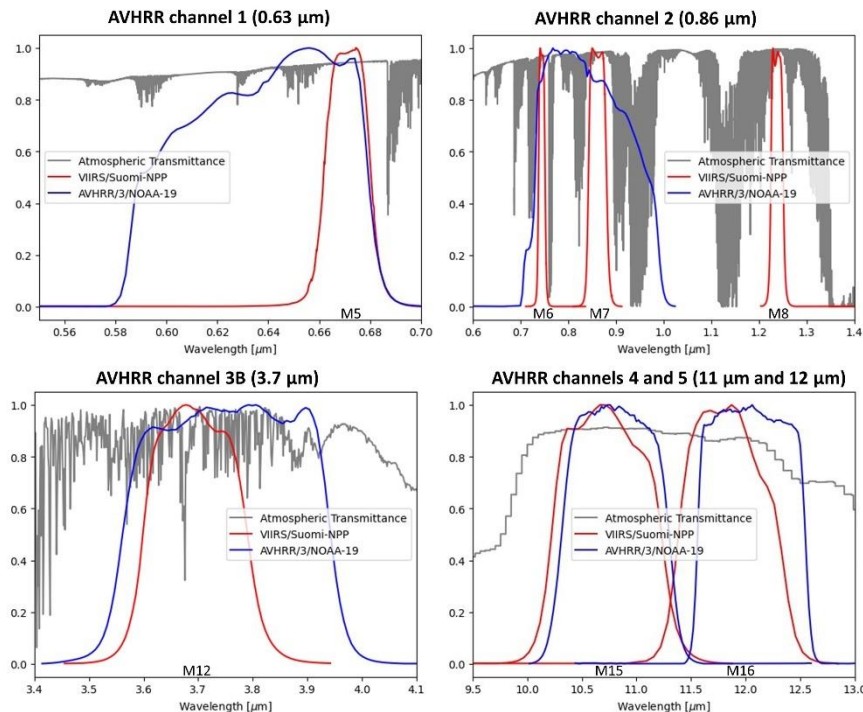

**Figure 1: Comparison of spectral responses for AVHRR (NOAA-19) and AVHRR-heritage channels of VIIRS (Suomi-NPP). Spectral response curves of AVHRR channels are given in blue to be compared with the response curves from VIIRS channels in red. Corresponding AVHRR-heritage channel notations (M5, M6, M7, M8, M12, M15, and M16) are provided at their central wavelengths along the x-axis. The grey curves give the atmospheric transmittance for reference.**

## 2.3 VGAC – Reduced resolution VIIRS data

The CLARA-A3 CDR is based on the archived global AVHRR dataset stored in a format called Global Area Coverage (GAC) with a horizontal resolution of approximately 4 km (Kidwell, 1991). Extending CLARA-A3 with VIIRS-derived products requires resampling VIIRS data to an equivalent horizontal resolution. This process benefits from a resampled VIIRS dataset already developed at NOAA (Knapp et al., 2019). This format is called VIIRS Global Area Coverage (VGAC), and VIIRS data in this format are currently available for almost the entire Suomi-NPP and some years of the NOAA-20 data record. The horizontal resolution is 3.9 km, and the resampling procedure (e.g., radiance averaging) is improved compared to the original GAC format for AVHRR. VGAC data have already been tested for use in CDR production (Wang et al., 2023; Seo et al., 2023). For this study, we used Suomi-NPP VGAC data from 2012, 2013, and 2019, as well as NOAA-20 VGAC data from 2019.

## 2.4 Selected approach

This study initially tested various SBAF relations, primarily sourced from NASA (2016). Results were acceptable for most AVHRR channels, but for some channels (especially channel 3B at 3.7 µm), we encountered problems in using the VIIRS-based simulations. For example, night-time cloud detection significantly overestimated low-level cloud amounts. The cloud detection method used (CMAPROB, described by Karlsson et al., 2020) is a probabilistic method using all AVHRR channels. AVHRR channel 3B is considered the most crucial channel for this method's performance, especially at night. Only minor deviations from the original AVHRR channel 3B radiances significantly affect the results at night. The encountered problems were likely caused by the limitation of IASI not observing radiances for wavelengths shorter than 3.62 µm. Since the AVHRR channel 3B and VIIRS band M12 spectral responses both allow for significant contributions at wavelengths shorter than 3.62 µm (see Fig. 1), this limitation can be substantial, especially since this affects in particular the contribution from reflected solar radiation which rapidly increases with decreasing wavelengths. An effort to describe these contributions using Radiative Transfer Model (RTM) calculations was applied in the satCORPS tool, but this was made using assumptions, making results more uncertain.

Due to the uncertainties encountered for the NASA-derived SBAFs for this channel, we decided to proceed by calculating SBAFs from collocated AVHRR- and VIIRS-observed radiances. In practice this means that the derived spectral band adjustments might be composed by more than just the effects of differences in spectral responses, since we cannot separate these effects from other effects (e.g. radiance biases or calibration errors) when doing collocations.

## 2.5 Selected collocation and validation datasets

Since VIIRS on the Suomi-NPP satellite was launched already in 2011 (with useful data delivered from January 2012 onwards), collocations with AVHRR measurements have been possible for more than a decade (Fig. 2). However, the only satellite allowing nearly simultaneous overpasses covering the entire globe was NOAA-19. In 2012 and 2013, NOAA-19 had nearly the same orbital configuration as Suomi-NPP, with a daytime equator crossing overpass time near 01:30 PM (i.e., local solar time 13:30 in Fig. 2). After 2013, the orbital drift of the NOAA-19 satellite gradually restricted available collocations to higher latitudes. Notice also that for later satellites carrying VIIRS (i.e., NOAA-20 launched in 2018 and NOAA-21 launched in 2022), these satellites still have fixed equator crossing times at 01:30 PM. Consequently, no global collocations with AVHRR were possible anymore.

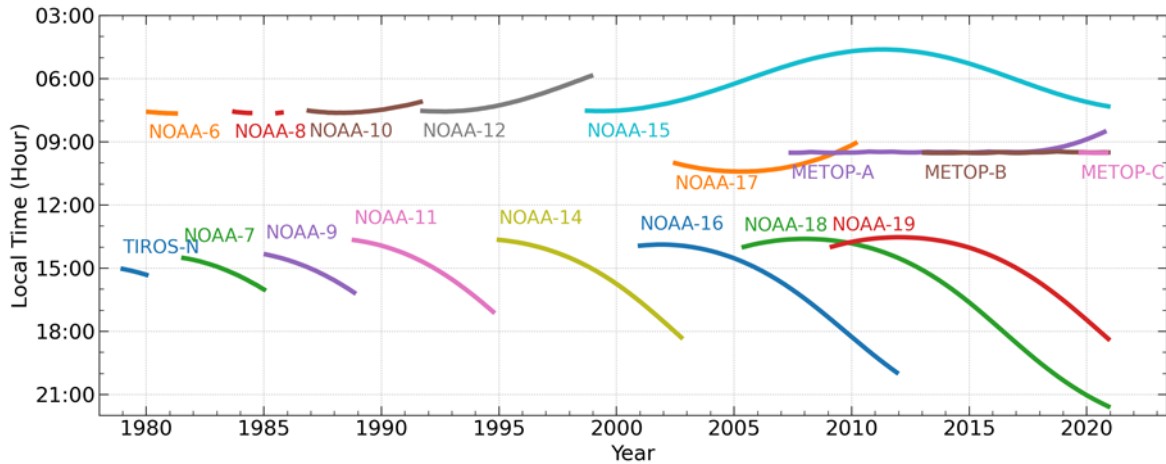

**Figure 2: Local solar times at equator observations for all AVHRR-carrying NOAA satellites from TIROS-N to NOAA-19 and EUMETSAT's METOP A/B/C satellites. Shown are all data that were used for the CLARA-A3 CDR processing. The figure shows ascending (northbound) equator crossing times for afternoon satellites (NOAA-7 to NOAA-19) and descending (southbound) equator crossing times for morning satellites (NOAA-12 to NOAA-17 and METOP A/B/C). Corresponding night-time observations take place 12 hours earlier/later. To be noticed is that Suomi-NPP has a stable orbit with equator crossing time at 13:30.**

Since NOAA-19 and Suomi-NPP had slightly different orbital altitudes and thus slightly different orbital periods, simultaneous observations (in this case, limited to within 2 minutes) occurred approximately every 15th day. Considering some data losses for both satellites, this resulted in a total collocation dataset of 115 Suomi-NPP orbits and corresponding NOAA-19 orbits for the two years. At most, pixel data were separated by 3 km, and maximum satellite zenith angles of up to 15 degrees to avoid too much influence from directional effects. Collocations were calculated using nearest neighbour matching with the Pyresample module in the Pytroll software package (Raspaud et al., 2018).

Table 3 lists the training and validation datasets used.

**Table 3: Description of the used training and validation datasets. The table shows the number of selected Suomi-NPP and NOAA-20 orbits being compared with the corresponding AVHRR radiances and cloud products, alternatively (for cloud product validation) the number of orbits being validated with CALIPSO cloud products. See text for details.**

| Dataset | Number of orbits | Period |
| --- | --- | --- |
| Training dataset 1 Suomi-NPP | 65 | 2012: Feb, Apr, Jun, Jul, Aug, Oct, Nov, Dec 2013: Jan, Mar, May, Jul, Sep, Nov |
| Training dataset 2 Suomi-NPP | 34 | 2013: Feb, Apr, Jun, Aug, Oct, Dec |
| Radiance validation dataset Suomi-NPP | 16 | 2012: Jan, Mar, May, Sep |
| Cloud product validation dataset Suomi-NPP | 289 | 2012, 2013, 2019: All months |
| Cloud product validation dataset NOAA-20 | 274 | 2019: All months |
| Cloud property inter-comparison dataset Suomi-NPP | 48 | 2012: All months |

The linear SBAF regression methods utilized the entire training dataset (dataset 1 and 2). For the Neural Network (NN) approach, training dataset 1 was used for the actual training and dataset 2 was used as a "during training validation dataset" to decide when to stop the training. The radiance validation dataset was used to evaluate the performance of all SBAF approaches. For the NN training, special handling of the AVHRR channel 3B for very low brightness temperatures (BT) had to be applied. The original AVHRR data have a rather poor radiometric resolution at very low temperatures, while the corresponding data

from the M12 channel of VIIRS are much improved compared to the AVHRR data. The poor radiometric resolution for AVHRR in this channel often led to missing data or a very large spread of realized BTs. It was decided not to let the network learn this type of irregular behaviour of AVHRR. Thus, data points with missing channel 3B data or data with BTs below 220 K were slightly modified using a simple relation between VIIRS and AVHRR BTs. Before training, the AVHRR BTs for these pixels were set to original VIIRS M12 BTs plus the difference between AVHRR Channel 4 and VIIRS M15 BTs, hereby assuming that the difference in BTs for 11 µm channels should be the same as the difference for 3.7 µm channels. This would stop the network from trying to learn these deficiencies of AVHRR Channel 3B for colder temperatures than 220 K.

In addition to validating the VGAC-simulated AVHRR radiances, inspecting the performance of certain CLARA products derived from these radiances proved necessary. As noted earlier, achieving a perfect simulation of AVHRR radiances is theoretically impossible. Therefore, it was important to check if the derived products were good enough to comply with CLARA product requirements. Thus, we produced the CLARA-A3 cloud products from the simulated AVHRR radiances. Comparisons with original NOAA-19 products (cloud property inter-comparison dataset in Table 3) and cloud datasets from the CALIPSO satellite (cloud product validation datasets in Table 3) were used to investigate them. More details on cloud product validations are given in Section 3.3.

## 3 Detailed description of applied spectral band adjustment methods and validation procedures

### 3.1 Spectral Band Adjustment (SBA) methods

We have tested two different SBA methods for the simulation of NOAA-19 AVHRR radiances from Suomi-NPP VIIRS radiances:

1. SBAs derived from linear regression
2. SBAs derived from a MultiLayer Perceptron (MLP) neural network

Method 1 is the classical regression method, often used in inter-calibration applications relating two nearby spectral channels, in which measurements can be collocated during an overlapping period. The two training data sets described in Table 3 were merged and used to estimate the regression parameters. The method has been applied with two different configurations:

Linear-1a:       Linear regression based on all training samples for individual channels

Linear-1b:       Linear regression separating results for day, night, and twilight

Configuration Linear-1b accounts for the fact some channels might behave differently during the day and night. This concern, in particular, AVHRR channel 3B at 3.7 µm, which measures exclusively thermally emitted radiation at night but

both thermally emitted and reflected solar radiation during the day. We have defined night as solar zenith angles (SZAs) above 89⁰, twilight for SZAs between 80-89⁰, and day as SZAs below 80⁰.

Appendix A provides linear regression parameters for both methods.

Method 2 (hereafter described as the NN-based method or SBA-NN) explores whether multichannel information from
VIIRS can be used to simulate individual AVHRR channel radiances, taking illumination conditions into account (in the same way as for method Linear-1b). The NN-based method was trained on all training samples, allowing dependence on multiple input channels for each target channel, separated by night, twilight, and daytime conditions. The following section further describes this method and its training.

### 3.2 Definition and training of the SBA-NN

This study used Multilayer perceptrons of the type Quantile Regression Neural Networks (QRNN, Pfreundshuh et al., 2018, Cannon, 2011). These have been successfully used to retrieve cloud top height parameters from polar satellite imagery (Håkansson et al., 2019) in the EUMETSAT Nowcasting Satellite Application Facility (NWC SAF) project. The resulting cloud top height products were also used later in CLARA-A3. Separate networks were trained for day, night, and twilight conditions. As input to the AVHRR training, the network had the individual channels (as reflectances and brightness
temperatures), channel differences (i.e., brightness temperature differences, BTDs), and channel ratios from VIIRS (e.g., M10 reflectance divided by M6 reflectance). As truth to train against, the same variables for AVHRR were used (Table 4). However, channels at 1.6 µm and 8.5 µm are not available on NOAA-19 AVHRR, which were consequently used only as input to the network. We omitted the variables concerning reflective channels during night-time conditions. The reason for using primarily channel differences and ratios between channels in the training, rather than just the original set of all individual channels, is
that many of the downstream applications for deriving CLARA products (e.g., cloud mask, cloud optical thickness, effective radii, and surface albedo) rely heavily on relations between two or more AVHRR channels. Thus, simulating these relations between channels as closely as possible is important.

The networks were trained for three distribution quantiles: of 16 %, 50 %, and 84 %, which estimate the retrieval error. The Sequential model form Keras/Tensorflow (Joseph et al., 2021) defined the networks, with the MLP settings detailed in Table
5. The code for the training and the resulting networks is available on Github (https://github.com/foua-pps/sbafs_ann). Table C1 in Appendix C provides the details on the final networks.

The data underwent thinning before training. The input data range was calculated and divided into 10 equal bins for each variable, from which 1,000 data points were randomly selected per bin. Bins with fewer than 1,000 points were supplemented with additional random selections to reach exactly 400,000 data points for the night-time network and 700,000 for each daytime
and twilight network. Data thinning aimed to give rare but important data points, such as cloud-free snow-covered surfaces and hot deserts, larger representation in the training data.

**Table 4: Training variables for the MLP network.**

| Variables |
| --- |
| Reflectance at 0.6 µm |
| Ratio of 0.6 µm and 0.9 µm reflectances |
| Brightness temperature at 11 µm |
| BTD between 12 µm and 11 µm |
| BTD between 3.7 µm and 11 µm |
| BTD between 8.5 µm and 11 µm |
| Ratio of 1.6 µm and 0.6 µm reflectances |

**Table 5: MLP network settings.**

| Multi-Layer Perceptron Model Setting | Value |
| --- | --- |
| Number of hidden layers | 2 |
| Number of neurons per hidden layer | 15 |
| Learning rate | 0.02 |
| Patience | 30 |
| Momentum | 0.9 |
| Decay | $10^{-6}$ |
| Activation function for hidden layers | Tanh |
| Kernel initializer | glorot uniform |

**3.3 Evaluation methods**

The results for the simulated radiances are primarily evaluated by inter-comparing radiances in scatter plots and calculating appropriate scores of radiance agreement. This evaluation is based on the Radiance validation dataset in Table 3.

An equally important way of evaluating the results is to investigate the impact on some central CLARA CDR cloud products, to verify that the products derived from simulated AVHRR radiances fulfilled essential CLARA-A3 CDR product requirements. A favourable condition for the chosen period was the possibility of collocating VIIRS and AVHRR orbits with observations from the CALIPSO satellite (also having equator crossing times near 01:30 PM). Consequently, cloud products

derived from VIIRS-simulated AVHRR data could be efficiently validated using CALIPSO-CALIOP cloud products (Winker, 2018). Thus, validation results for the cloud parameters cloud fractional cover (CFC in %), cloud top height (CTH, in meters) and cloud phase (CPH, meaning the percentage of liquid phase cloud tops) could be derived. The validation methods were the same as those described by Karlsson and Håkansson (2018).

The processing of CLARA CDR cloud products is based on the software package for the Polar Platform System (PPS), originally developed within the EUMETSAT NWC SAF project (https://www.nwcsaf.org/web/guest/home). PPS enables cloud product processing for a wide range of imagers. The shift from one sensor to another is generally dealt with by adjusting pre-calculated cloud detection thresholds, atmospheric corrections and other adaptations from mainly RTTOV-simulations utilizing each sensors' spectral response functions.

We used a variety of validation scores depending on the investigated parameters (all scores described in detail by Karlsson and Håkansson, 2018). For the cloud mask evaluation, we first converted cloud probabilities to a binary cloud mask using the probability threshold of 50 %. Then, we calculated the overall cloud fractional cover (CFC) to be compared with the CFC calculated from the CALIPSO cloud products. Validation scores mean error (bias in percentage points - %), dimensionless Hit Rate and Kuipers scores were then calculated. Since the CALIOP cloud lidar on CALIPSO is more sensitive to cloud occurrence than AVHRR, the Kuipers and Hit Rate scores were also calculated for CALIOP observations with the thinnest clouds excluded (i.e., excluding contributions from cloud layers with cloud optical thicknesses < 0.20). For CTH, the cloud top of the topmost cloud layer detected by CALIOP was used as the validation reference. Computed validation scores for CTH were mean error and mean absolute error (both in meters). Also here, the effect of removing the thinnest clouds from the CALIPSO dataset was studied, but now using a slightly relaxed cloud optical thickness threshold of 0.4, claiming that the prospect of retrieving a cloud top height requires that clouds are also detected with reasonable confidence.

Notice that the Suomi-NPP/CALIPSO matchups for the years 2012-2013 (i.e., basically the Cloud product validation dataset for Suomi-NPP in Table 3) are independent of the AVHRR/Suomi-NPP matchups (i.e., the two training datasets for Suomi-NPP) in the same period. More clearly, the likelihood of simultaneous nadir observations from all three platforms is extremely low.

In addition to evaluating the performance of cloud products for 2012 and 2013, we also made the same validation effort based on CALIPSO data for all months in 2019. The purpose was mainly to choose a period with more independent data, well separated in time from the two years when SBAF methods were initially derived. Thus, we wanted to see if the results were still valid for Suomi-NPP data 6 years later. In addition, we also wanted to see if the next VIIRS sensor on NOAA-20, with slightly different spectral responses for some involved channels, would produce results with similar quality. This validation effort used data from cloud validation datasets from 2019, as shown in Table 3.

As a final confirmation that simulated CLARA-A3 cloud products would perform satisfactorily overall, we also made inter-comparisons between original and simulated AVHRR cloud property products for the NOAA-19/Suomi-NPP matchups in the year 2012. Cloud property products COT, CRE, LWP, and IWP could not be validated in the CALIPSO validation study. Comparing them with original AVHRR products indicates their validity, even though the validation dataset (i.e., the Cloud

property inter-comparison dataset in Table 3) partly includes cases used for training the methods. The analysis also computes agreement scores similar to those defined for the radiance inter-comparisons.

## 4 Results

### 4.1 Results for simulated AVHRR radiances

Figure 3 shows scatterplots with results of the simulated AVHRR reflectances for the two visible AVHRR channels (Channels 1 and 2, see Table 1). For the linear regressions, we only show results of version Linear-1b, since the two linear versions give almost identical results for the visible channels. Results are compared with original reflectances from AVHRR and VIIRS. It is clear that original channel reflectances are highly correlated in both sensors, although VIIRS reflectances are generally slightly higher than AVHRR reflectances. The relatively large spread around the identity line is largely explained by remaining,

but small collocation errors in space and time. Furthermore, sampling differences in how radiances are averaged in AVHRR GAC and VIIRS VGAC Field Ff Views (FOVs) can also contribute to the scatter.

Simulations with the linear method improves the agreement (i.e., brings results closer to the identity line) in Fig. 3 for both channels, but appear to create overcompensated results for high reflectances. The NN-based method delivers the best results, effectively removing most of the off-diagonal deviations observed in the linear method.

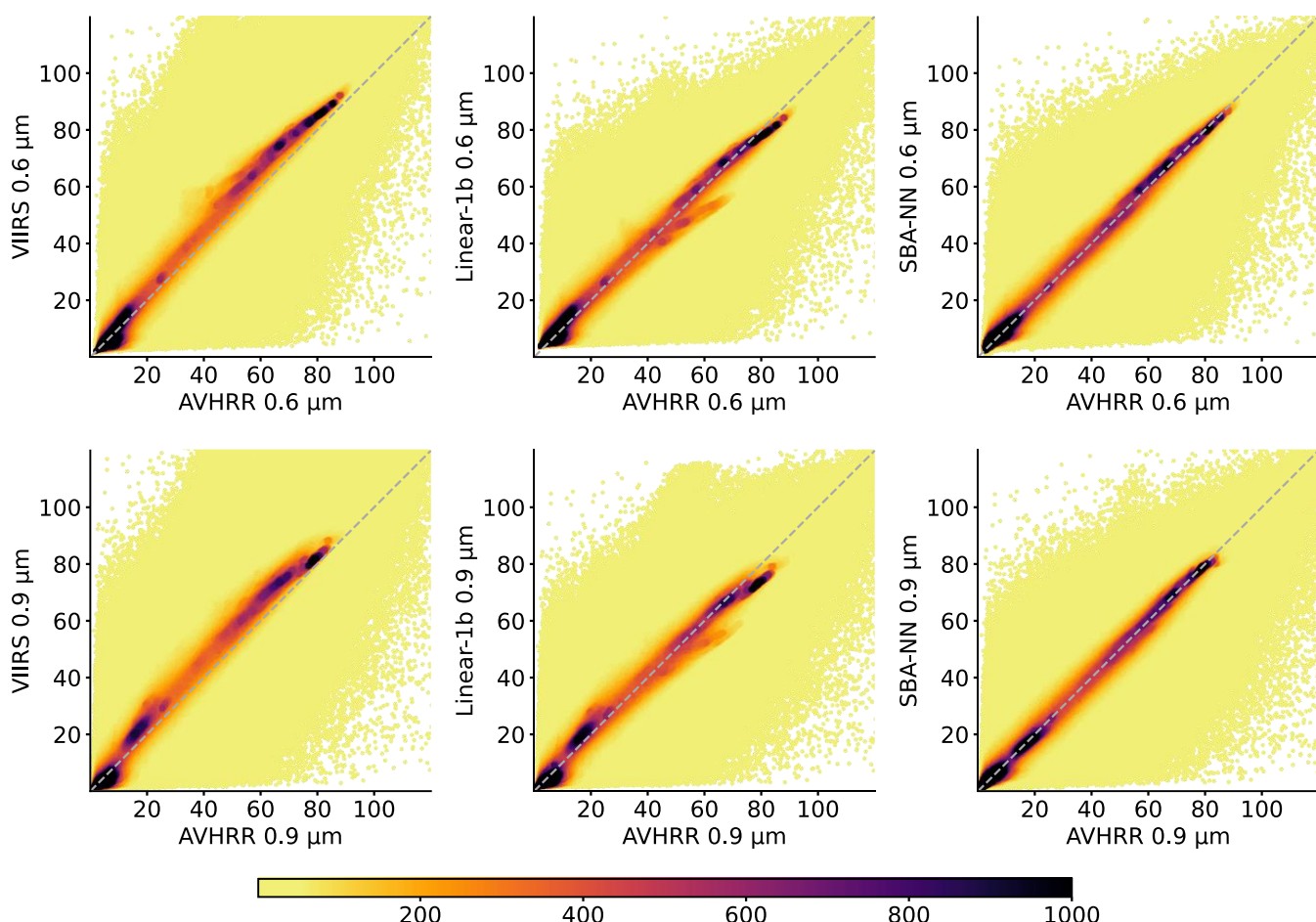

**Figure 3: Radiance inter-comparisons for AVHRR channels 1 (0.6 μm, upper panels) and 2 (0.9 μm, bottom panels). Original AVHRR vs VIIRS reflectances (%) are shown in the leftmost column. AVHRR vs VIIRS-simulated reflectances are shown by the central and rightmost columns. The center column shows results from the Linear-1b method, while the rightmost column shows simulated results from the SBA-NN method.**

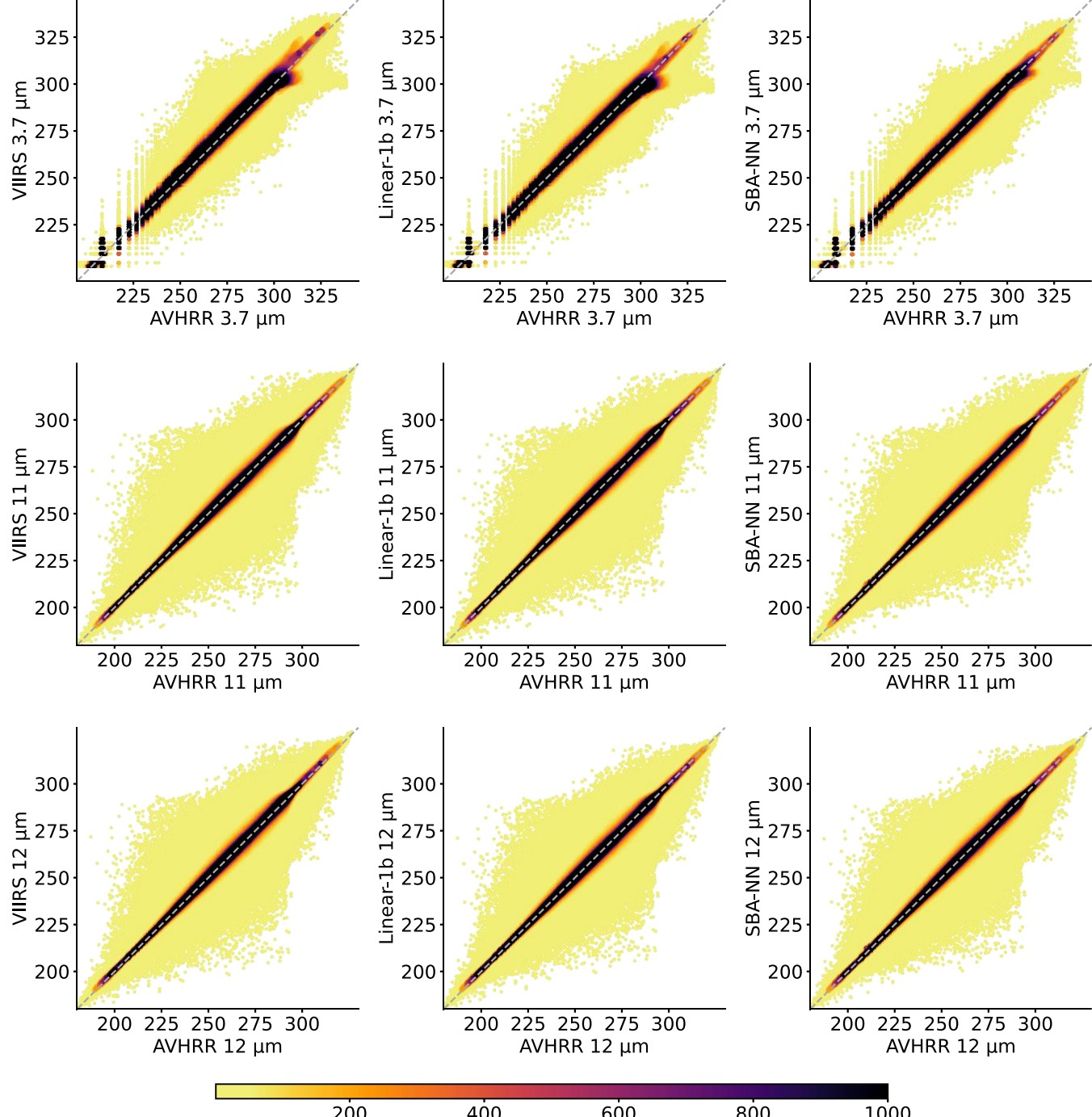

**Figure 4: Radiance inter-comparisons for AVHRR channel 3b (3.7 μm, uppermost panels), channel 4 (11 μm, middle panels) and channel 5 (12 μm, lowermost panels). The figure shows merged results for all daytime categories. Original AVHRR vs VIIRS brightness temperatures (K) are shown in the leftmost column. AVHRR vs VIIRS-simulated**

**brightness temperatures are shown by the central and rightmost columns. The center column shows results from the Linear-1b method, while the rightmost column shows simulated results from the SBA-NN method.**

As seen in Fig. 1, the spectral responses for VIIRS and AVHRR for the infrared channels 4 and 5 are very similar, which is also verified by the results in the middle and bottom panels of Fig. 4. Original and simulated results for the two AVHRR channels are here more or less identical. Some deviations are, however, seen for AVHRR channel 3B (upper panel in Fig. 4) and the NN-based method is somewhat better than the linear method for reducing these deviations.

Although AVHRR simulations for the infrared and shortwave-infrared channels appear to agree well with the original AVHRR radiances in Fig. 4, we must emphasize that for some CLARA applications (like cloud detection), the quality of simulated AVHRR channel differences is very important. Similarly, estimating surface albedo in CLARA requires accurate inter-channel relations (i.e., reflectance ratios) between the two visible channels. Fig. 5 closes in on those features.

The close-up of differences and ratios in Fig. 5 reveals deviations from the identity line for both original and simulated results. The brightness temperature differences for the three original infrared and short-wave infrared channels are relatively large. The linear approach somewhat improves the agreement, but the NN-based method can almost completely remove the differences. The linear method cannot improve the agreement for the reflectance ratio, while the NN-based method provides an excellent agreement.

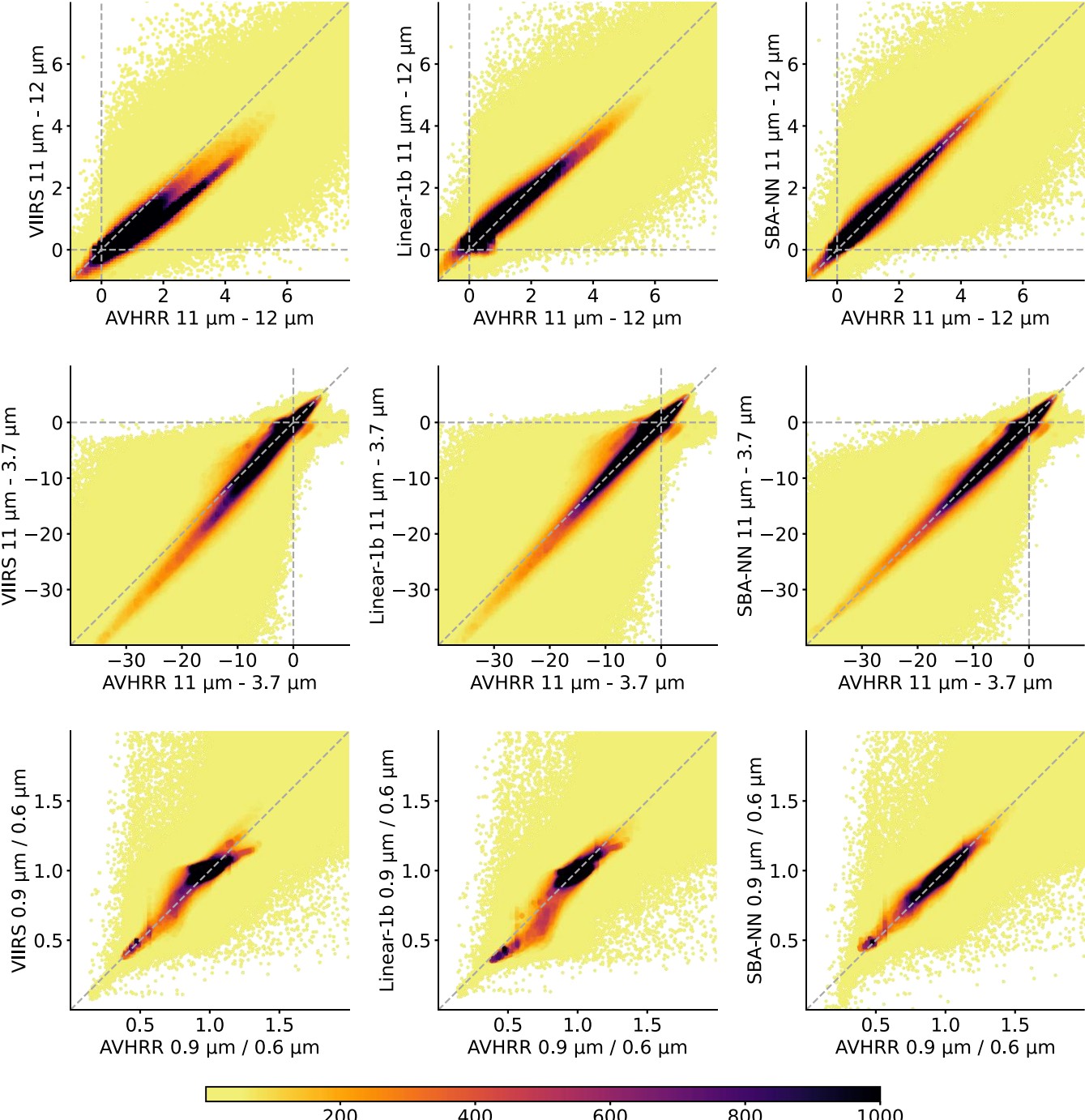

**Figure 5: Results for selected channel differences and ratios. The figure shows merged results for all three daytime categories. Uppermost panels: Results for original and simulated brightness temperature differences (K) between AVHRR channels 4 and 5 (11-12 µm). Middle panels: Results for original and simulated brightness temperature**

**differences (K) between AVHRR channels 4 and 3B (11-3.7 μm). Lowermost panels: Results for original and simulated reflectance ratio between AVHRR channels 2 and 1 (0.9/0.6 μm). Original AVHRR and VIIRS relations are shown in the leftmost column, results for the Linear-1b method in the middle column, and results for the NN-based method in the rightmost column.**


Figure 6 summarises the quality (uncertainty) of simulated results, i.e., the difference distribution between simulated and real AVHRR results for the studied methods based on the same datasets as in Figs. 3, 4, and 5. Notice that the figure also shows results for the previously illustrated channel differences and ratios, and results for the Linear-1a method (not shown in Figs. 4 and 5). For reference, the figure also shows the original deviation between AVHRR and VIIRS channels.

We immediately notice that the SBA-NN method is superior to all other methods by showing the highest frequency at the zero-difference level (blue curves in Fig. 6). This method creates no secondary peaks, and the distribution is clearly non-Gaussian (i.e., the grey distribution indicated in Fig. 6 is Gaussian with the same bias and standard deviation). Notice, in particular, this method has excellent results for channel differences and ratios (lowest row in Fig. 6).

Results in Fig. 6 clearly show that AVHRR channel 3B simulations require separate handling of day and night measurements.

If the simulation method is not different between day and night (i.e., as in Linear-1a), the error difference clearly peaks outside the zero level (e.g., green curves in the leftmost plots in the middle and bottom rows in Fig. 6).

Tables A2.1-A2.4 in Appendix B summarize all results for selected overall statistical measures. Appendix B also provides results sub-divided for day, night, and twilight conditions.

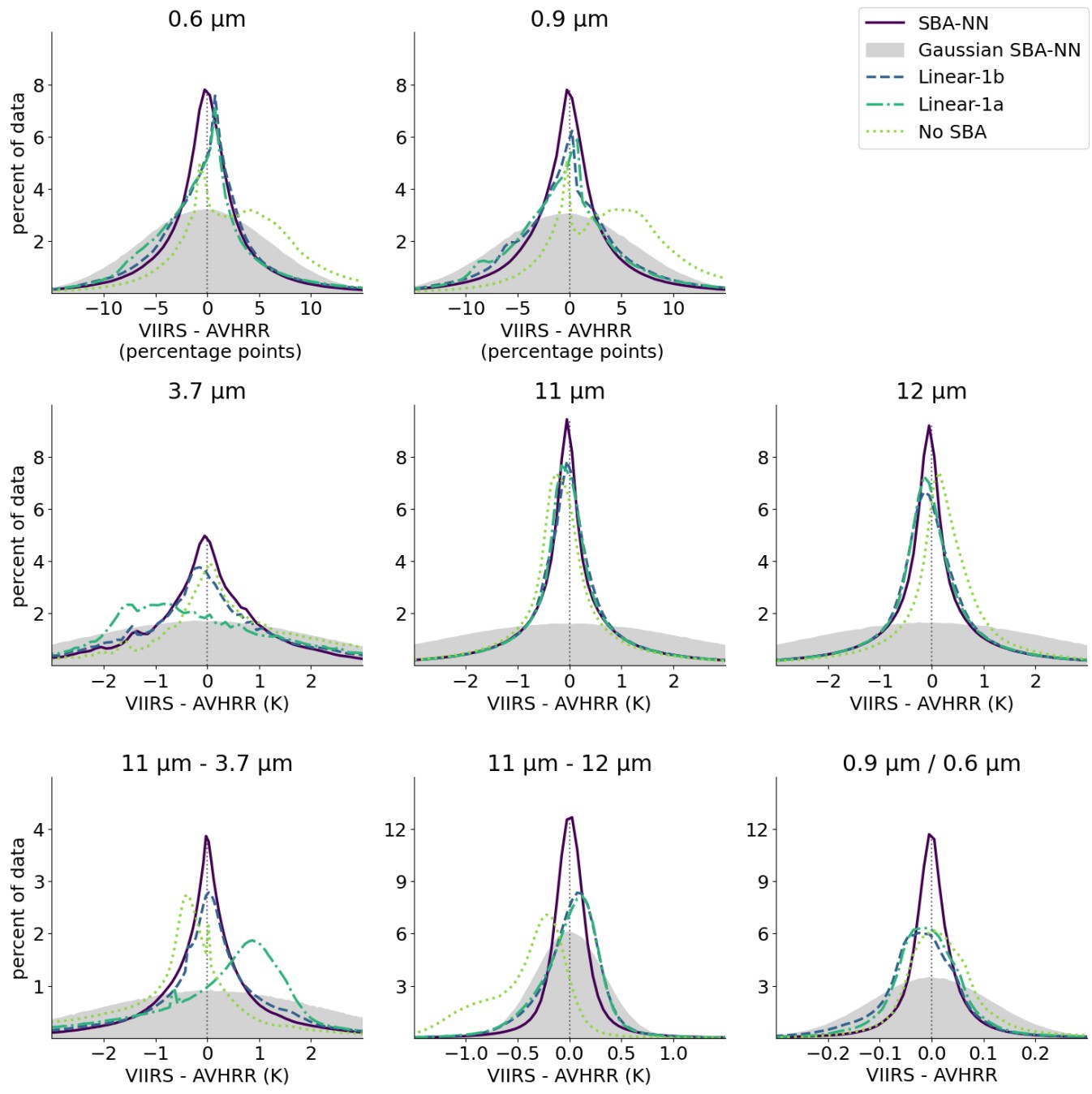


**Figure 6: Frequency distributions of differences between simulated and real AVHRR radiances and brightness temperatures. For reference, also the original AVHRR and VIIRS channel deviations are shown (green dotted curve = No SBA in the legend). The upper two panels show results for all 5 AVHRR channels, while the bottom panels show**

**results for BTDs (channel 4 – channel 5) and channel 4 – channel 3B) and reflectance ratio (channel 2 divided by channel 1).**

## 4.2 CALIPSO-based validation of derived CLARA cloud products for 2012 and 2013

Tables 6-8 show the results from the CALIPSO-CALIOP validation of three CLARA-A3 cloud products based on
VIIRS/VGAC-simulated AVHRR radiances. Section 3.3 describes the general validation setup and the validation scores used. The tables contain validation results for eight different validation setups:

1. **NOAA-19 CLARA-A3**

   Achieved validation results in a previous validation of CLARA-A3 products for products generated from the NOAA-
19 AVHRR instrument over 2012-2013. These results are included, since this study aims to produce results from VIIRS/VGAC data compatible with earlier NOAA-19 AVHRR-based products.

2. **PPS VIIRS**

   Validation results based on uncorrected AVHRR-heritage channels of VIIRS/VGAC data in 2012-2013. Notice that this category means that PPS cloud products have been produced in the VIIRS environment (i.e., applying pre-
calculated cloud detection thresholds, atmospheric corrections and other adaptations using spectral responses of the five AVHRR-heritage channels of VIIRS).

3. **VGAC No SBA**

   Similar to the previous category but now based on PPS cloud products produced in the AVHRR environment (i.e., applying pre-calculated cloud detection thresholds, atmospheric corrections and other adaptations using spectral
responses of the original five NOAA-19 AVHRR channels). This environment is also used for all following categories.

4. **VGAC Linear-1a**

   Validation results based on VIIRS/VGAC-simulation of AVHRR channels based on the method Linear-1a in 2012-2013.

5. **VGAC Linear-1b**

   Validation results based on VIIRS/VGAC-simulation of AVHRR channels based on the method Linear-1b in 2012-2013.

6. **VGAC SBA-NN**

   Validation results based on VIIRS/VGAC-simulation of AVHRR channels based on the SBA-NN method in 2012-
430   2013.

7. **VGAC SNPP 2019 SBA-NN**

Validation results based on Suomi-NPP VIIRS/VGAC-simulation of AVHRR channels based on the SBA-NN method in 2019.

8. **VGAC NOAA-20 2019 SBA-NN**

Validation results based on NOAA-20 VIIRS/VGAC-simulation of AVHRR channels based on the SBA-NN method in 2019.

The tables also show the original requirements for the three cloud products in the CLARA-A3 CDR in the rightmost columns. Products generated from VIIRS/VGAC-simulated data should also fulfil these requirements. In the CLARA-A3 evaluation,
very thin clouds detected by CALIPSO-CALIOP were removed based on COT thresholds of 0.2 and 0.4 for CFC and CTH, respectively. Karlsson and Håkansson (2018) suggested this CFC COT threshold after studying the effect of COT thresholding during the CLARA-A2 CDR CFC validation exercise. They found the best overall validation scores when excluding clouds with COT lower than this threshold. This COT-thresholding was later used to define the CLARA-A3 CFC requirement (Table 6, rightmost column). Here, using different CFC and COT thresholds (Tables 6 and 7) is motivated by wanting to discard the
thinnest clouds when validating the CTH product. These clouds are always the most difficult to deal with for any CTH retrieval, i.e., the thinner the clouds, the more challenging it becomes to compensate for semi-transparency effects. The COT threshold for CTH validation is more arbitrarily chosen. It should be higher than 0.2, although not drastically, since cloud detection efficiency increases rapidly for COTs larger than 0.2 (see Karlsson and Håkansson, 2018). A reasonable threshold of 0.4 was found to remove some of these thin cloud uncertainties. The threshold should not be too high to give justice to all semi-
transparency correction efforts, i.e., it would not make sense to only look at opaque clouds. To highlight improvements more clearly, Tables 6 and 7 show both the COT-thresholded results, used for CLARA-A3 requirements, and the original results that include optically thin clouds. Relying only on the COT-thresholded results would overlook important improvements to the overall CDR, particularly for the CFC product.

**Table 6. Validation scores for Cloud Fractional Cover (CFC): bias [%], Kuipers score, and hitrate (both [-]). See text and Sect. 3.3 for details.**

| Parameter | NOAA-19 CLARA-A3 | PPS VIIRS | VGAC No SBA | VGAC Linear-1a | VGAC Linear-1b | VGAC SBA-NN | VGAC SNPP 2019 SBA-NN | VGAC NOAA-20 2019 SBA-NN | CLARA-A3 Product require-ment |
|---|---|---|---|---|---|---|---|---|---|
| Total # Orbits | 1026 | 497 | 497 | 497 | 497 | 497 | 289 | 274 | |
| Total # FOVs | 6 355 780 | 3 468 354 | 3 468 354 | 3 468 354 | 3 468 354 | 3 468 351 | 1 891 597 | 1 804 134 | |
| **CFC** bias | -11.03 % | -10.46 % | -7.47 % | -6.11 % | -10.20 % | -9.74 % | -10.37 % | -9.43 % | - |
| **CFC** bias (COT > 0.2) | -0.22 % | 0.28 % | 3.27 % | 4.63 % | 0.55 % | 1.00 % | 0.67 % | 1.63 % | 5 % |
| **CFC** Kuipers | 0.687 | 0.701 | 0.639 | 0.606 | 0.688 | 0.695 | 0.694 | 0.687 | - |
| **CFC** Kuipers | 0.706 | 0.712 | 0.646 | 0.638 | 0.710 | 0.700 | 0.701 | 0.685 | 0.6 |

| | | | | | | | | |
|---|---|---|---|---|---|---|---|---|
| (COT > 0.2) | | | | | | | | |
| **CFC** Hitrate | 0.825 | 0.833 | 0.822 | 0.814 | 0.829 | 0.834 | 0.831 | 0.833 | - |

**Table 7. Validation scores for Cloud Top Height (CTH): bias and mean absolute error (both [m]). Total number of used orbits and samples are given in Table 6. See text and Sect. 3.3 for details.**

| Parameter | NOAA-19 CLARA-A3 | PPS VIIRS | VGAC No SBA | VGAC Linear-1a | VGAC Linear-1b | VGAC SBA-NN | VGAC SNPP 2019 SBA-NN | VGAC NOAA-20 2019 SBA-NN | CLARA-A3 Product require-ment |
|---|---|---|---|---|---|---|---|---|---|
| **CTH** bias | -900 m | -1049 m | -1129 m | -504 m | -501 m | -598 m | -633 m | -408 m | - |
| **CTH** bias (COT > 0.4) | 807 m | 644 m | 641 m | 1166 m | 1143 m | 1034 m | 1122 m | 1288 m | 800 m |
| **CTH** mean abs error | 1664 m | 1678 m | 1755 m | 1560 m | 1538 m | 1541 m | 1673 m | 1656 m | - |

**Table 8. Validation scores for Cloud Phase (CPH, here the fraction of liquid clouds): bias [%], Kuipers score, and hitrate (both [-]). Total number of used orbits and samples are given in Table 6. See text and Sect. 3.3 for details.**

| Parameter | NOAA-19 CLARA-A3 | PPS VIIRS | VGAC No SBA | VGAC Linear-1a | VGAC Linear-1b | VGAC SBA-NN | VGAC SNPP 2019 SBA-NN | VGAC NOAA-20 2019 SBA-NN | CLARA-A3 Product require-ment |
|---|---|---|---|---|---|---|---|---|---|
| **CPH** mean bias | -1 % | -4 % | -6 % | -1 % | 1 % | 0 % | 0 % | 2 % | 5 % |
| **CPH** Kuipers | 0.67 | 0.66 | 0.66 | 0.69 | 0.68 | 0.68 | 0.67 | 0.68 | 0.6 |
| **CPH** Hitrate | 0.84 | 0.83 | 0.83 | 0.85 | 0.85 | 0.84 | 0.84 | 0.84 | - |

If comparing results in columns "PPS VIIRS" and "VGAC No SBA" for CFC in Table 6, we notice that to process VIIRS data for AVHRR-heritage channels in the correct PPS VIIRS environment generally gives better results than when processing these data in the AVHRR PPS environment. It proves that there is indeed a need to apply spectral band adjustments to properly use VIIRS-based data in the AVHRR PPS environment. Some signs of this is also seen for CPH results in Table 8 while it is more difficult to judge the changes seen for CTH results in Table 7.

Regarding CFC results, it is clear that all methods (except Linear-1a which actually worsens results further) perform well, indeed much better than the CLARA-A3 requirements and very close to the achieved validation results for CLARA-A3. The SBA-NN method has the best overall scores for the VIIRS/VGAC simulations validated against all CALIPSO-detected clouds. A closer look at results (not shown here) reveals that clearly better results are mainly explained by superior CFC performance during night conditions.

Results for CTH in Table 7 are also somewhat similar for all methods, but it is difficult to draw conclusions based exclusively on the bias parameter. As Håkansson et al. (2018) pointed out, the cloud top distribution is largely bi-modal with peaks for low-level and high-level clouds. Since determining high-level cloud tops is much more difficult than for low-level cloud tops (i.e., high-level clouds are predominantly semi-transparent), the actual distribution of low-level and high-level clouds in the validation dataset has therefore great importance for the bias results. Consequently, the error structure is generally non-Gaussian, making the bias parameter inappropriate as a measure of uncertainty. A more reliable uncertainty parameter here is the mean absolute error. We notice that all results based on simulated AVHRR data are in line with or even slightly better than the AVHRR reference results from CLARA-A3.

For the CPH product results in Table 8, the differences in the results between the various products are even smaller, so no method can be determined as clearly standing out compared to any other method.

It is encouraging is to see that the method found to best simulate AVHRR radiances according to the previous section (the SBAF-NN method), also appears to perform well based on Suomi-NPP and NOAA-20 data from 2019. Only a minor degradation appears visible for NOAA-20, which has a VIIRS sensor with slightly different spectral channel responses compared to Suomi-NPP.

## 4.3 Inter-comparing original AVHRR-based and VIIRS/VGAC simulated cloud physical products for 2012

Only some of the CLARA-A3 cloud products can be validated using CALIPSO-CALIOP. The cloud physical products (CPP) of CLARA-A3 consist not only of the CPH product (validated in the previous section), but also of the CWP product, sub-divided into cloud liquid water path (LWP) and ice water path (IWP). COT and CRE are needed for their generation. Even if we cannot easily validate these products from independent data, comparing the original AVHRR-based products with the VIIRS/VGAC-generated products in the used AVHRR/VIIRS collocation dataset is possible. Here, we show such inter-comparisons based on all collocated data for 2012. Notice that since the CPP products are only derived during daytime, the only difference between the Linear-1a and Linear-1b methods is that the latter also has a distinction between twilight and day.

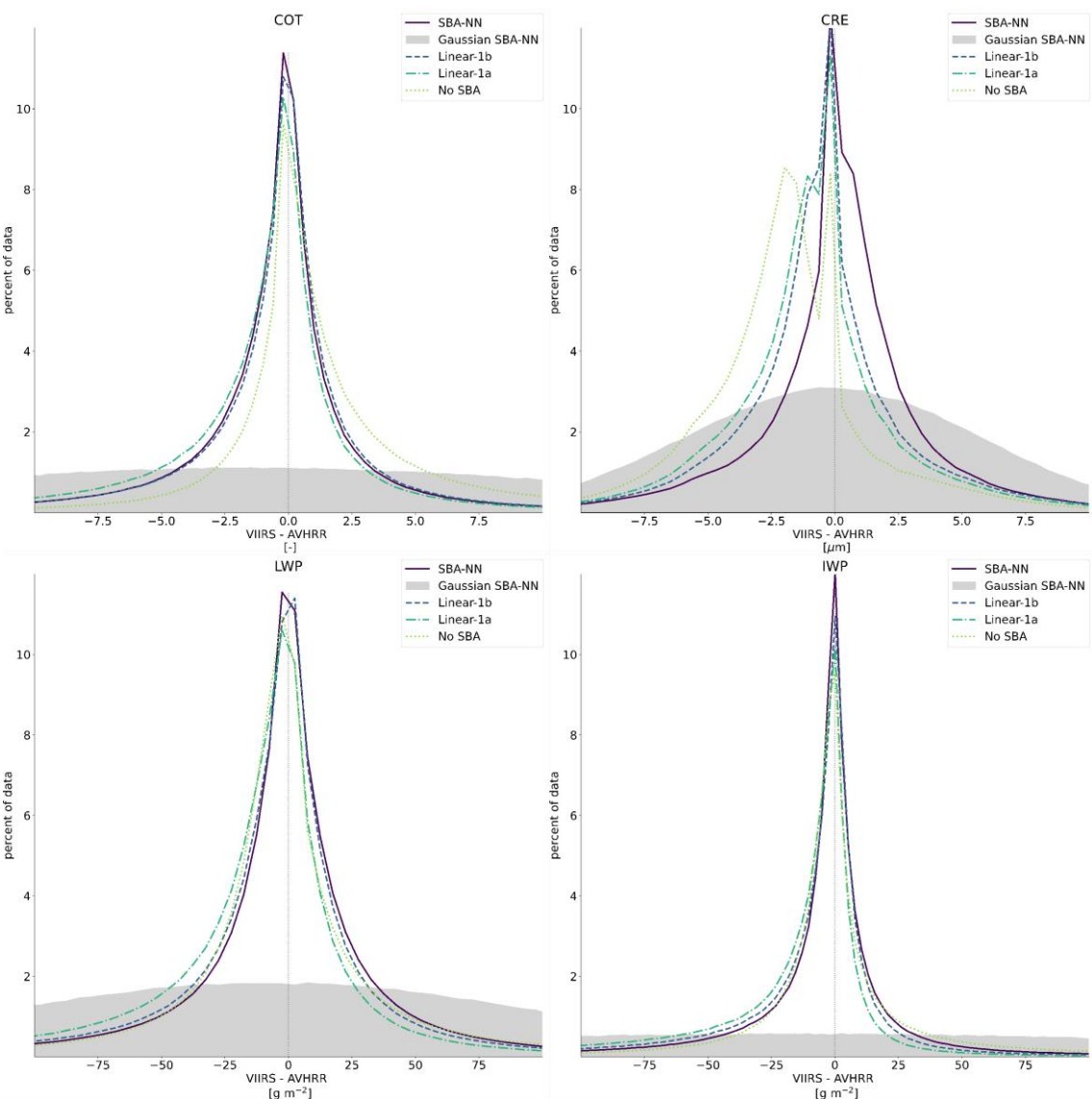

**Figure 7: Frequency distributions of CPP differences between products based on VIIRS-simulated and real AVHRR radiances. For reference, also products based on uncorrected VIIRS radiances are shown (green dotted curve = No SBA in the legend).**

**Table 9. Mean Absolute Difference (MAD) for the various Spectral Band Adjustment methods for CPP products based on simulated data versus products based on original AVHRR data. Minimum MADs for each CPP product are shown in bold numbers.**

| | COT [-] | CRE [μm] | LWP [gm$^{-2}$] | IWP [gm$^{-2}$] |
|---|---|---|---|---|
| **No SBA** | 8.8 | 3.4 | 61.1 | 85.8 |
| **Linear-1a** | 5.9 | 3.0 | 49.8 | 75.9 |
| **Linear-1b** | 5.4 | 3.0 | 44.8 | 67.8 |
| **SBA-NN** | **5.1** | **2.6** | **42.0** | **55.9** |

Figure 7 shows frequency distributions of the difference between VIIRS-simulated CPP products and original AVHRR-based CPP products for all SBA-methods. The figure also shows resulting frequencies for the case when no spectral band adjustments are applied (i.e., the No SBA case). Table 9 summarizes the resulting mean absolute deviations (MAD) for the different cases.

It is clear from Fig. 7 that if no SBA-corrections are made, the difference distributions of the two fundamental parameters COT and CRE (i.e., fundamental for the calculation of LWP and IWP) are not symmetrically distributed around the zero-difference value. Especially the CRE difference distribution appears biased with a secondary peak for negative differences (i.e., underestimated CREs compared to the AVHRR values). SBA corrections improve the results and especially the SBA-NN method produce a well-defined narrow peak centered at the zero-difference level. This is also reflected in Table 9 showing minimum MADs for the SBA-NN method.

## 5 Discussion

### 5.1 AVHRR radiance simulations

Our simulations show that radiance simulations based on linear regression methods can only partially remove differences between original AVHRR and AVHRR-heritage VIIRS channels. Although correlations between channels are generally high (especially for the infrared channels), there are obvious remaining non-linear features that linear methods cannot handle properly. In contrast, the SBA-NN method handles these non-linear features more accurately when simulating radiances. Including channel differences and ratios during network training, which is particularly important for downstream applications, further improves the results.

One very important finding in our study is that for the 3.7 µm channel (AVHRR channel 3B), which measures both emitted thermal and reflected solar radiation during daytime, it is crucial to separate radiance simulations for daytime and night-time conditions. It is well-known that radiances in this channel behave very differently during night and day. However, it is also clear that small differences in spectral responses between the original AVHRR channel 3B and the AVHRR-heritage channel M12 of VIIRS lead to clearly different daytime behaviour of the two channels. This is a consequence of the small difference in central wavelengths shown in Tables 1 and 2. The lower central wavelength value in M12 means that this channel is slightly more sensitive to reflected solar radiation, which results in a larger reflectance contribution to the observed brightness temperature in this channel. Thus, any relation deduced from purely night-time measurements (with only thermal emissions) will fail if applied to daytime measurements. From this, we also need to emphasize that, even if previously shown results for cloud screening in Table 6 indicate that good results could be achieved without any spectral band adjustments of VIIRS heritage (i.e., staying with VIIRS processing of AVHRR-heritage channels without any adaptations to AVHRR data) channels, any retrieval that is highly dependent on AVHRR channel 3B need to make spectral band adjustments on the corresponding VIIRS channel.

Central to this study was the definition of the AVHRR/VIIRS collocation dataset for 2012 and 2013. The fact that NOAA-19 and Suomi-NPP had very similar, but not identical, orbits caused some concern, since it led to an uneven global sample extraction. The small orbital differences led to the largest orbital track separations in the tropics, resulting in large (less favourable) angular differences. To avoid too much influence of angular differences, which could lead to differences from anisotropic reflection and atmospheric absorption effects, we set a maximum viewing angle of 15 degrees for both sensors. This value was a compromise between the wish of getting samples for all latitudes, but still limiting the effects of differences due to anisotropic reflection and atmospheric absorption. It allowed samples to be obtained from all latitudes, but the number of resulting samples also became an increasing function of latitude. Initially, we tried even stricter limits on viewing angles, but this resulted in only a few samples from very warm surfaces at low latitudes, thereby hampering the neural network training.

### 5.2 Resulting cloud masks, cloud top heights, and cloud phases

The CLARA-A3 CDR's cloud detection method depends heavily on radiances in AVHRR channel 3B. As mentioned in the previous section, the difficulties in handling this channel properly clearly affected validation results based on CALIPSO-

CALIOP cloud products. Using the radiance simulation method Linear-1a resulted in serious errors in night-time cloud masking, even if daytime results were acceptable. Night-time results clearly improved with the Linear-1b method, but even more so with the NN-based method.

The performance of the SBA-NN cloud mask was even slightly better than the original results from NOAA-19 in the CLARA-A3 CDR. However, it should be noticed that the previous validation of CLARA-A3 products allowed slightly larger time

differences between AVHRR and CALIPSO observations (i.e., 5 minutes instead of 2 minutes) which probably led to slightly degraded results for CLARA-A3 compared to using a maximum 2 minutes time difference in both datasets.

Results for the other two CALIPSO-examined cloud products did not show a large variation and the existing variation mostly was linked to the cloud mask quality.

Results for these three cloud products, based on the different AVHRR simulations and on original AVHRR and uncorrected

VIIRS radiances, generally did not differ very much in the achieved validation results (at least, if not including night-time results for the method Linear-1a). In fact, they all reside well within the requirements previously set up for CLARA-A3. We conclude that, for these products, the AVHRR-heritage channels of VIIRS, corrected or not, contain enough information to provide results close to original AVHRR-based products. Encouraging is also that the good results are stable enough to be repeated in 2019 for the Suomi-NPP satellite. Results are also good for the VIIRS sensor on NOAA-20 in 2019, although with

some degradation compared to Suomi-NPP. We expect this degradation to some extent due to differences in spectral responses between the two VIIRS sensors. However, the differences remain small enough to avoid the need for radiance corrections, as the results still meet CLARA-A3 product requirements.

## 5.3 Resulting cloud physical parameters

The inter-comparison of original AVHRR-based CPP products and products based on VIIRS-simulated radiances showed that very accurate AVHRR radiance simulations are more important here than for the previously discussed cloud products. This concerns the COT parameter (which depends largely on AVHRR channel 1) and the CRE parameter (which depends largely on AVHRR channel 3B). A simulation of visible and short-wave infrared radiances should preferably be sub-divided into twilight and daytime categories. However, the most important for the simulation of channel 3B is that the simulation method

must be derived exclusively from day-time data.

The success of the SBA-NN method compared to the linear methods reveals something very interesting: The neural network appears capable of applying different radiance correction factors to different objects in simulated channel 3B images. Notice that only objects able to reflect solar radiation in channel 3B should have a correction factor that differs from the one applied during night-time (when only thermally emitted radiation is measured). Furthermore, this correction factor should also vary

depending on the reflection efficiency of the object, i.e., the correction factor should be dependent on the object's reflectance. Any linear correction method will fail here, since the correction factor is practically always constant, regardless of the object

in the image. The objects reflecting the most in this channel are typically thick water clouds. Since the linear regression will produce a nearly constant average correction factor (only slightly moderated by an offset parameter), the correction factor in this channel is likely too small for thick water clouds and too large for darker objects with low reflectance (e.g., snow-covered

surfaces). This is one explanation for underestimating simulated liquid cloud CRE values, which leads to underestimated LWP values. Only the SBA-NN method can correct for this effect.

Figure 8 and Table 10 illustrate together the capability of the SBAF-NN method to generate different SBAs in AVHRR channel 3B for different objects in a VIIRS/VGAC-simulation of AVHRR data over the Greenland area from 21 July 2012. Crosses mark locations in the scene with different cloud and surface types as interpreted by the Polar Platform System (PPS) cloud

type product (Dybbroe et al. 2004). Table 10 shows the resulting SBA corrections for AVHRR channel 3B for the Linear-1b and the SBA-NN methods in all the marked points. Especially, notice the different SBA corrections for SBA-NN for highly reflecting objects ($X_2$, $X_4$ and $X_7$) compared to those for weakly reflecting or non-reflecting objects ($X_3$, $X_5$ and $X_6$). It means that the NN is capable of identifying the different behaviour of different objects, thus making a kind of implicit object type identification when assigning which SBAs to use.

These results show the advantage of the NN approach in absorbing information, not only from the closest AVHRR-heritage channel, but from a larger set of VIIRS channels for making it possible to give different objects different effective SBA corrections. For example, notice that snow surfaces, which should hardly be SBA-corrected at all according to Table 10, are known to have very low reflectance in AVHRR channels 3A and 3B. Thus, when including also channel M10 in the training process, this could help in identifying snow surfaces with higher confidence than if using solely channel M12 in this spectral

region. Another example is the inclusion of VIIRS channel M14 in the training process, a channel which is not available at all on AVHRR. This channel is partly affected by water vapour absorption and this might potentially be useful for getting a better treatment of the simulation of brightness temperature differences between AVHRR channels 4 and 5, generally affected by both water vapour absorption and differences in cloud transmissivities.


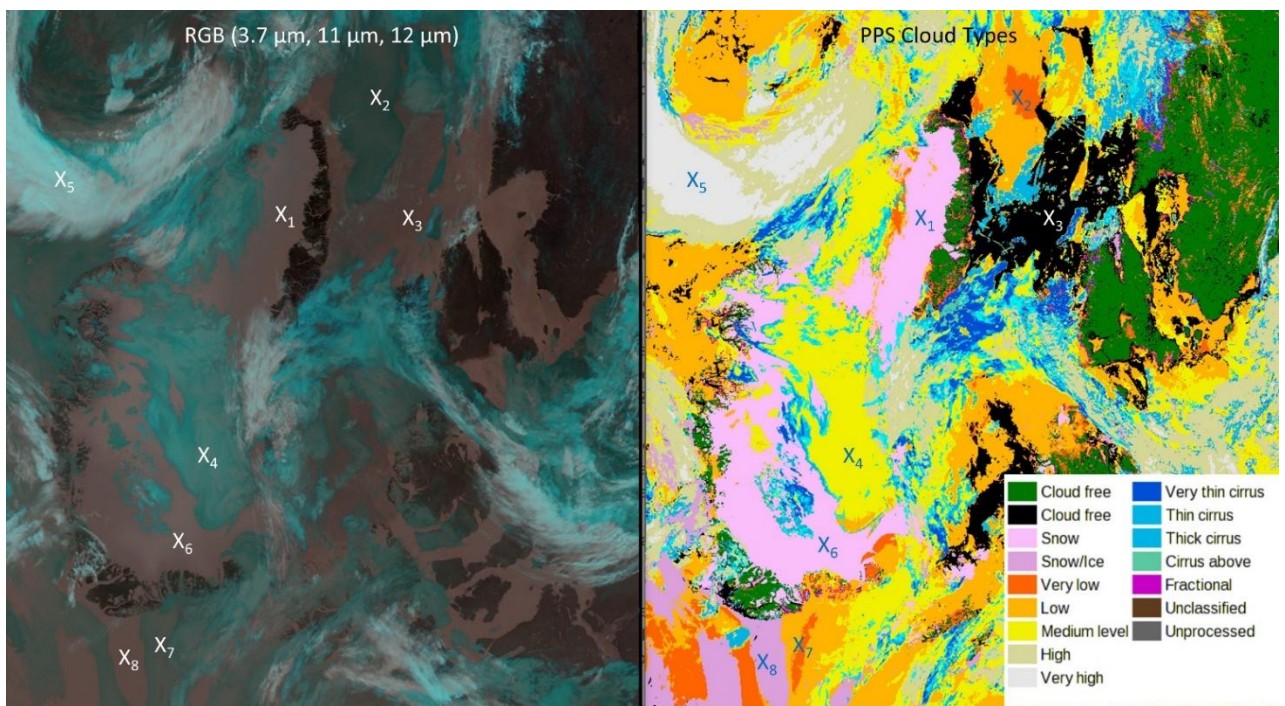

**Figure 8: Excerpt of a VIIRS/VGAC-simulated AVHRR orbit from Suomi-NPP 21 July 2012 (with first orbit scanline at 15:16 UTC). The scene shows Greenland, the Labrador Sea, and adjacent Canadian islands (with southern direction upwards) in a colour composite (left) based on VIIRS-simulated AVHRR channels 3B, 4, and 5, and a corresponding PPS cloud type classification (right). SBAF corrections for the Linear-1b and SBA-NN methods for the marked positions are given in Table 10.**

**Table 10: Resulting SBA corrections (i.e., adjustment factor compared to uncorrected VIIRS channels) for simulation methods Linear-1b and SBA-NN for markup points in Fig. 8.**

| Markup point and associated cloud or surface type | SBAF Linear-1b 3.7 µm | SBA-NN 3.7 µm |
|---|---|---|
| $X_1$: Snow (wet) | 0.993 | 0.999 |
| $X_2$: Very low (Stratus) | 0.992 | 0.987 |
| $X_3$: Cloud-free ocean | 0.992 | 1.000 |
| $X_4$: Medium level clouds | 0.992 | 0.985 |
| $X_5$: Very high clouds | 0.992 | 1.000 |

| | | |
|---|---|---|
| X$_6$: Snow (dry) | 0.993 | 0.999 |
| X$_7$: Low (Stratocumulus) | 0.993 | 0.992 |
| X$_8$: Ice-covered ocean | 0.993 | 0.998 |


### 5.4 Prospects for remaining CLARA-A3 products

We have shown that NOAA-19 AVHRR radiances can be simulated from Suomi-NPP VIIRS/VGAC radiances and that they can be used to produce cloud products with the same or even better quality than the original products included in the CLARA-A3 CDR. Whether the other CLARA-A3 products (i.e., surface albedo, surface radiation, and TOA radiation) can also be reproduced successfully from VIIRS/VGAC data remains to be investigated. Preliminary tests based on the SBA-NN radiance simulations have been performed for all products with promising results. However, more extensive validations need to take place, which is outside this paper's scope. Nevertheless, there is no reason to believe in the failure of the remaining products if basic radiances and cloud products are produced with high quality.

### 6 Conclusions

AVHRR radiances from the NOAA-19 satellite have been successfully simulated from Suomi-NPP VIIRS/VGAC radiances in 2012-2013, when observations from the two satellites could be collocated efficiently in both time and space. Two methods based on linear regression for each channel and one method based on an MLP neural network have been tested. The latter was trained using all AVHRR-heritage channels on VIIRS plus a few additional channels. Special attention was given to day and night differences, while constraints on channel differences and channel ratios were applied to the neural network.

The neural network approach achieved the best results for all individual channels. We found it crucial to separate daytime from night-time results when simulating AVHRR channel 3B at 3.7 µm. The small spectral response differences between AVHRR channel 3B and the VIIRS M12 channel, leading to a smaller effective wavelength for the VIIRS channel, meant that different simulation methods had to be used for day and night. Furthermore, radiance corrections during daytime had to depend on the actual object reflectance to be realistic. Only the neural network approach was able to achieve this.

Selected NOAA-19 cloud products (cloud mask, cloud top height, and cloud phase) in the CLARA-A3 CDR were produced from the VIIRS/VGAC-simulated radiances based on the same retrieval methods as used when compiling CLARA-A3. The resulting products were validated using CALIPSO-CALIOP cloud products and compared to the original CLARA-A3 products. Product qualities agreed well with original CLARA-A3 products and are clearly within product requirements. In addition, the cloud microphysical products COT, CRE, LWP, and IWP were well reproduced when based on the neural network-simulated radiances.

In order to check the validity of the derived spectral adjustments on a longer time scale, it is suggested to regularly repeat the validation efforts on derived cloud products based on new observations from active sensors on EarthCARE (Illingworth et al.,

2015) and similar missions in the future. In addition, regular checks of Simultaneous Nadir Observations (SNOs) at high

latitudes between VIIRS and IASI (and its successors) can help in deducing the infrared bands spectral responses' stability. The possibility to make similar checks for VIIRS and METimage visible bands seems unfortunately not possible from the Tropospheric Ozone Monitoring Instrument (TROPOMI, Veefkind et al., 2012), due to very limited spectral coverage of the AVHRR-heritage channels. The Ocean Color Instrument (OCI) on the PACE mission (Plankton, Aerosol, Cloud, ocean Ecosystem; Gorman et al., 2019) and the EMIT sensor (Earth Mineral dust source Investigation) aboard the International Space

Station offer potential sources of suitable reference measurements. Sentinel-3's Ocean and Land Colour Instrument (OLCI; Donlon et al., 2012) can also provide reference data. While OLCI is not fully hyperspectral, it includes 21 bands within the 0.4–1.0 µm range and operates with a 10:00 AM equator overpass time.

CLARA-A3 will be complemented and extended with the VIIRS/VGAC-based products to cover the period 1979-2022 (44 years) and this edition will be named CLARA-A3.5. Interestingly, for the future, the same radiance simulation approach could

be applied on radiances from the upcoming METimage sensor on the EPS-SG satellites, with an expected first launch in August 2025. This satellite will have nearly the same equator crossing time as the current Metop-C satellite with the last AVHRR sensor onboard. This would enable collocations in the same way as the currently used collocations of NOAA-19 and Suomi-NPP data. If AVHRR-radiances can be successfully simulated from METimage data, the CLARA CDR can be extended by several decades based on measurements from VIIRS and METimage sensors. Finally, the possibility to also use MODIS-based

AVHRR data simulations can be considered to improve observational coverage for some earlier years in the CLARA CDR (e.g., for 1999-2001 when the NOAA-16 orbital drift was considerable). However, highest priority is to secure a CDR extension with VIIRS and METimage data.

**Appendix A: Coefficients for linear regression methods**

**Table A1: Linear regression parameters for method Linear-1a based on all data (i.e., merged training datasets 1 and 2 in Table 3). For details on channel wavelengths for the two sensors, see Table 1 and Table 2.**

| AVHRR Channel | Simulated from VIIRS channel | Slope | Offset | Number of observations |
|---|---|---|---|---|
| Channel 1 (%) | M5 | 0.8534 | 1.8517 | 19,010,195 |
| Channel 2 (%) | M7 | 0.8507 | 1.1157 | 19,010,195 |
| Channel 3B (K) | M12 | 0.9734 | 6.1707 | 41,474,568 |
| Channel 4 (K) | M15 | 1.0006 | -0.0378 | 41,474,568 |
| Channel 5 (K) | M16 | 0.9906 | 2.1505 | 41,474,568 |

**Table A2: Linear regression parameters for method Linear-1b based on all data (i.e., merged training datasets 1 and 2 in Table 3) but restricted to DAY conditions. For details on channel wavelengths for the two sensors, see Table 1 and Table 2.**

| AVHRR Channel | Simulated from VIIRS channel | Slope | Offset | Number of observations |
|---|---|---|---|---|
| Channel 1 (%) | M5 | 0.8960 | 1.7118 | 15,102,309 |
| Channel 2 (%) | M7 | 0.8907 | 0.5547 | 15,102,309 |
| Channel 3B (K) | M12 | 0.9817 | 3.0744 | 15,102,309 |
| Channel 4 (K) | M15 | 0.9926 | 2.0934 | 15,102,309 |
| Channel 5 (K) | M16 | 0.9774 | 5.6555 | 15,102,309 |

**Table A3: Linear regression parameters for method Linear-1b based on all data (i.e., merged training datasets 1 and 2 in Table 3) but restricted to TWILIGHT conditions. For details on channel wavelengths for the two sensors, see Table 1 and Table 2.**

| AVHRR Channel | Simulated from VIIRS channel | Slope | Offset | Number of observations |
|---|---|---|---|---|
| Channel 1 (%) | M5 | 0.6710 | 4.5075 | 3,904,507 |
| Channel 2 (%) | M7 | 0.7120 | 3.7473 | 3,904,507 |
| Channel 3B (K) | M12 | 0.9973 | -0.7479 | 3,904,507 |
| Channel 4 (K) | M15 | 1.0024 | -0.5164 | 3,904,507 |
| Channel 5 (K) | M16 | 0.9967 | 0.6742 | 3,904,507 |


**Table A4: Linear regression parameters for method Linear-1b based on all data (i.e., merged training datasets 1 and 2 in Table 3) but restricted to NIGHT conditions. For details on channel wavelengths for the two sensors, see Table 1 and Table 2.**

| AVHRR Channel | Simulated from VIIRS channel | Slope | Offset | Number of observations |
|---|---|---|---|---|
| Channel 3B (K) | M12 | 0.9948 | 1.3254 | 21,971,492 |
| Channel 4 (K) | M15 | 1.0042 | -0.9186 | 21,971,492 |
| Channel 5 (K) | M16 | 0.9962 | 0.7373 | 21,971,492 |


**Appendix B: Detailed score statistics for radiance and image feature simulations**

Table B1: Radiance validation scores (for quantity VIIRS – AVHRR) for DAY (4 307 482 samples). Shown are values
of median error (median), mean absolute error (MAE), interquartile range (IQR), mean error (bias),root-mean-
squared error (RMSE) and regression parameters (Slope, Offset and Correlation) for the simulation of each AVHRR
channel and each channel combination, and for every tested simulation method. Individual best scores are highlighted
in bold numbers.

| Channel or feature | Method | median | MAE | IQR | bias | RMS | Slope | Offset | Corre-lation |
|---|---|---|---|---|---|---|---|---|---|
| 0.6 μm (%) | SBA-NN | **-0.091** | **3.700** | **3.878** | -0.219 | **6.331** | 0.961 | 1.157 | **0.970** |
| | Linear-1b | 0.310 | 3.941 | 4.652 | **0.052** | 6.444 | 0.945 | 1.969 | 0.969 |
| | Linear-1a | -0.905 | 4.278 | 5.394 | -1.401 | 6.675 | 0.901 | 2.096 | 0.969 |
| | No-correction | -1.926 | 5.107 | 6.840 | 2.228 | 7.500 | 1.055 | 0.287 | 0.969 |
| 0.9 μm (%) | SBA-NN | **-0.134** | **3.817** | **4.060** | -0.290 | **6.464** | 0.953 | 1.341 | **0.968** |
| | Linear-1b | -0.175 | 4.403 | 5.605 | **-0.110** | 6.904 | 0.933 | 2.229 | 0.963 |
| | Linear-1a | -0.774 | 4.646 | 5.988 | -1.094 | 7.054 | 0.891 | 2.715 | 0.963 |
| | No-correction | 3.318 | 5.929 | 7.690 | 3.557 | 8.398 | 1.048 | 1.880 | 0.963 |
| 3.7 μm (K) | SBA-NN | **0.008** | **1.779** | **2.093** | -0.122 | **2.899** | 0.972 | 7.802 | **0.987** |
| | Linear-1b | -0.180 | 2.617 | 3.582 | -0.343 | 3.805 | 0.933 | 18.523 | 0.978 |
| | Linear-1a | 0.606 | 2.694 | 3.646 | 0.384 | 3.829 | 0.926 | 21.488 | 0.978 |
| | No correcction | 1.820 | 3.096 | 3.520 | 1.806 | 4.176 | 0.951 | 15.736 | 0.978 |
| 11 μm (K) | SBA-NN | -0.070 | 1.528 | 1.024 | -0.008 | **3.371** | 0.983 | 4.711 | 0.986 |
| | Linear-1b | -0.072 | 1.564 | 1.114 | -0.035 | 3.378 | 0.973 | 7.279 | 0.986 |
| | Linear-1a | **-0.043** | **1.521** | **0.977** | **-0.007** | 3.382 | 0.981 | 5.190 | 0.986 |
| | No correction | -0.170 | 1.542 | 0.980 | -0.131 | 3.384 | 0.980 | 5.224 | 0.986 |
| 12 μm (K) | SBA-NN | -0.073 | **1.504** | 1.000 | **-0.001** | **3.311** | 0.982 | 4.908 | 0.986 |
| | Linear-1b | -0.134 | 1.557 | 1.160 | -0.035 | 3.324 | 0.972 | 7.499 | 0.986 |
| | Linear-1a | **-0.065** | 1.512 | **0.991** | 0.011 | 3.334 | 0.985 | 4.019 | 0.986 |
| | No correction | 0.290 | 1.596 | 1.060 | 0.389 | 3.377 | 0.994 | 1.886 | 0.986 |
| 0.9 μm/0.6 μm (-) | SBA-NN | **-0.000** | **0.054** | **0.051** | **-0.001** | **0.123** | 0.896 | 0.104 | **0.952** |
| | Linear-1b | -0.024 | 0.079 | 0.088 | -0.022 | 0.147 | 1.013 | -0.035 | 0.942 |
| | Linear-1a | -0.001 | 0.068 | 0.082 | 0.007 | 0.139 | 0.976 | 0.031 | 0.943 |
| | No-correction | 0.026 | 0.096 | 0.088 | 0.062 | 0.237 | 1.299 | -0.239 | 0.937 |
| 11 μm – 12 μm (K) | SBA-NN | **-0.001** | **0.217** | **0.230** | -0.007 | **0.395** | 0.910 | 0.124 | **0.947** |
| | Linear-1b | 0.032 | 0.266 | 0.344 | **-0.001** | 0.426 | 0.817 | 0.265 | 0.941 |
| | Linear-1a | 0.030 | 0.279 | 0.385 | -0.018 | 0.434 | 0.794 | 0.280 | 0.941 |
| | No correction | -0.420 | 0.566 | 0.620 | -0.520 | 0.732 | 0.739 | -0.141 | 0.915 |
| 11 μm – 3.7 μm (K) | SBA-NN | **-0.049** | **1.981** | **2.165** | 0.114 | **3.272** | 0.908 | -1.143 | **0.940** |
| | Linear-1b | 0.255 | 2.754 | 3.444 | 0.308 | 4.127 | 1.060 | 1.120 | 0.928 |

| | | | | | | | | | |
|---|---|---|---|---|---|---|---|---|---|
| | Linear-1a | -0.554 | 2.879 | 3.657 | -0.391 | 4.213 | 1.065 | 0.498 | 0.926 |
| | No-correction | -1.890 | 3.343 | 3.530 | -1.937 | 4.575 | 1.070 | -0.992 | 0.929 |

**Table B2: Radiance validation scores (for quantity VIIRS – AVHRR) for TWILIGHT (852 881 samples). Shown are values of median error (median), mean absolute error (MAE), interquartile range (IQR), mean error (bias),root-mean-squared error (RMSE) and regression parameters (Slope, Offset and Correlation) for the simulation of each AVHRR channel and each channel combination, and for every tested simulation method. Individual best scores are highlighted in bold numbers.**

| Channel or feature | Method | median | MAE | IQR | bias | RMS | Slope | Offset | Corre-lation |
|---|---|---|---|---|---|---|---|---|---|
| 0.6 μm (%) | SBA-NN | **-0.191** | **3.902** | **5.275** | -0.289 | **5.832** | 0.893 | 3.996 | **0.932** |
| | Linear-1b | -1.280 | 6.778 | 9.636 | **0.128** | 9.355 | 0.705 | 11.981 | 0.813 |
| | Linear-1a | 4.246 | 8.273 | 9.519 | 7.205 | 12.654 | 0.897 | 11.356 | 0.813 |
| | No-correction | 10.386 | 13.618 | 10.982 | 13.177 | 17.863 | 1.051 | 11.137 | 0.813 |
| 0.9 μm (%) | SBA-NN | **-0.312** | **4.490** | **5.865** | -0.490 | **7.164** | 0.880 | 4.642 | **0.927** |
| | Linear-1b | -0.894 | 7.233 | 10.089 | **0.215** | 10.395 | 0.752 | 10.774 | 0.839 |
| | Linear-1a | 2.568 | 7.739 | 9.445 | 5.192 | 12.374 | 0.899 | 9.512 | 0.839 |
| | No-correction | 9.207 | 13.059 | 11.397 | 12.267 | 17.911 | 1.056 | 9.869 | 0.839 |
| 3.7 μm (K) | SBA-NN | **0.132** | **0.972** | **1.222** | **0.129** | **1.518** | 0.977 | 6.114 | **0.990** |
| | Linear-1b | -0.223 | 1.458 | 2.227 | 0.137 | 2.010 | 0.980 | 5.213 | 0.983 |
| | Linear-1a | 0.569 | 1.533 | 2.226 | 0.886 | 2.190 | 0.957 | 11.988 | 0.983 |
| | No correction | 1.210 | 1.879 | 2.230 | 1.582 | 2.556 | 0.983 | 5.977 | 0.983 |
| 11 μm (K) | SBA-NN | 0.028 | 0.707 | 0.697 | 0.033 | 1.285 | 0.991 | 2.248 | 0.996 |
| | Linear-1b | **0.018** | **0.658** | **0.574** | **0.012** | **1.257** | 0.992 | 1.890 | 0.996 |
| | Linear-1a | 0.052 | 0.663 | 0.585 | 0.041 | 1.258 | 0.991 | 2.365 | 0.996 |
| | No correction | -0.060 | 0.665 | 0.590 | -0.071 | 1.259 | 0.990 | 2-401 | 0.996 |
| 12 μm (K) | SBA-NN | **0.019** | 0.712 | 0.694 | 0.028 | 1.299 | 0.991 | 2.293 | 0.996 |
| | Linear-1b | -0.026 | **0.671** | 0.599 | **-0.010** | **1.276** | 0.992 | 1.879 | 0.996 |
| | Linear-1a | -0.070 | 0.691 | 0.670 | -0.054 | 1.279 | 0.986 | 3.348 | 0.996 |
| | No correction | 0.120 | 0.682 | **0.580** | 0.138 | 1.284 | 0.996 | 1.209 | 0.996 |
| 0.9 μm/0.6 μm (-) | SBA-NN | **-0.003** | **0.041** | **0.053** | **-0.004** | **0.063** | 0.832 | 0.170 | **0.916** |
| | Linear-1b | 0.005 | 0.054 | 0.080 | 0.005 | 0.076 | 0.660 | 0.357 | 0.886 |
| | Linear-1a | -0.050 | 0.069 | 0.079 | -0.051 | 0.091 | 0.690 | 0.269 | 0.883 |
| | No-correction | -0.032 | 0.060 | 0.079 | -0.031 | 0.082 | 0.703 | 0.277 | 0.876 |
| 11 μm – 12 μm (K) | SBA-NN | **0.008** | **0.138** | **0.194** | **0.005** | **0.213** | 0.910 | 0.068 | **0.957** |
| | Linear-1b | 0.039 | 0.158 | 0.224 | 0.022 | 0.229 | 0.852 | 0.126 | 0.952 |
| | Linear-1a | 0.115 | 0.189 | 0.235 | 0.095 | 0.253 | 0.860 | 0.193 | 0.948 |
| | No correction | -0.190 | 0.241 | 0.250 | -0.209 | 0.320 | 0.836 | -0.094 | 0.946 |
| 11 μm – 3.7 μm (K) | SBA-NN | **-0.101** | **0.873** | **1.057** | -0.096 | **1.410** | 0.971 | -0.297 | **0.981** |
| | Linear-1b | 0.293 | 1.412 | 2.163 | **-0.125** | 1.946 | 1.130 | 0.761 | 0.979 |
| | Linear-1a | -0.466 | 1.441 | 2.131 | -0.845 | 2.106 | 1.128 | 0.033 | 0.979 |
| | No-correction | -1.240 | 1.867 | 2.140 | -1.653 | 2.544 | 1.127 | -0.786 | 0.979 |

**Table B3: Radiance validation scores (for quantity VIIRS – AVHRR) for NIGHT (5 758 113 samples). Shown are values of median error (median), mean absolute error (MAE), interquartile range (IQR), mean error (bias), root-mean-squared error (RMSE) and regression parameters (Slope, Offset and Correlation) for the simulation of each AVHRR**

 **channel and each channel combination, and for every tested simulation method. Individual best scores are highlighted in bold numbers.**

| Channel or feature | Method | median | MAE | IQR | bias | RMS | Slope | Offset | Correlation |
|---|---|---|---|---|---|---|---|---|---|
| **3.7 μm (K)** | SBA-NN | -0.069 | 1.164 | **1.208** | -0.085 | **1.988** | 0.995 | 1.215 | 0.996 |
| | Linear-1b | -0.073 | 1.185 | 1.255 | **-0.013** | 1.998 | 0.992 | 2.127 | 0.996 |
| | Linear-1a | -0.902 | **1.554** | 1.672 | -0.768 | 2.205 | 0.970 | 6.955 | 0.996 |
| | No correction | **0.010** | 1.178 | 1.240 | 0.022 | 2.000 | 0.997 | 0.805 | 0.996 |
| **11 μm (K)** | SBA-NN | -0.038 | 0.925 | 0.788 | -0.042 | 1.772 | 0.996 | 1.050 | 0.997 |
| | Linear-1b | **-0.020** | 0.925 | 0.788 | **-0.000** | 1.772 | 0.995 | 1.332 | 0.997 |
| | Linear-1a | -0.080 | 0.945 | 0.834 | -0.054 | 1.775 | 0.991 | 2.205 | 0.997 |
| | No correction | -0.200 | 0.971 | 0.850 | -0.172 | 1.783 | 0.991 | 2.241 | 0.997 |
| **12 μm (K)** | SBA-NN | -0.046 | **0.905** | **0.773** | -0.052 | **1.732** | 0.996 | 1.064 | 0.997 |
| | Linear-1b | **-0.044** | 0.939 | 0.874 | **-0.007** | 1.748 | 0.996 | 0.963 | 0.997 |
| | Linear-1a | -0.086 | 0.956 | 0.930 | -0.043 | 1.750 | 0.991 | 2.374 | 0.997 |
| | No correction | 0.190 | 0.970 | 0.890 | 0.239 | 1.768 | 1.000 | 0.226 | 0.997 |
| **11 μm – 12 μm (K)** | SBA-NN | **0.007** | **0.172** | **0.216** | 0.009 | **0.281** | 0.954 | 0.061 | **0.972** |
| | Linear-1b | 0.052 | 0.248 | 0.368 | **0.007** | 0.343 | 0.829 | 0.197 | 0.964 |
| | Linear-1a | 0.028 | 0.245 | 0.368 | -0.010 | 0.343 | 0.838 | 0.170 | 0.963 |
| | No correction | -0.320 | 0.443 | 0.480 | -0.411 | 0.581 | 0.779 | -0.166 | 0.948 |
| **11 μm – 3.7 μm (K)** | SBA-NN | **0.023** | **0.615** | **0.646** | 0.043 | **1.071** | 0.925 | -0.100 | **0.959** |
| | Linear-1b | 0.027 | 0.640 | 0.708 | **0.013** | 1.091 | 0.950 | -0.081 | 0.958 |
| | Linear-1a | 0.830 | 1.066 | 1.006 | 0.714 | 1.379 | 0.952 | 0.622 | 0.951 |
| | No-correction | -0.270 | 0.706 | 0.730 | -0.194 | 1.122 | 0.944 | -0.300 | 0.957 |

**Table B4: Radiance validation scores (for quantity VIIRS – AVHRR) for ALL cases (10 918 476 samples but for visible channels and features 5 160 363 samples). Shown are values of median error (median), mean absolute error (MAE), interquartile range (IQR), mean error (bias), root-mean-squared error (RMSE) and regression parameters (Slope, Offset and Correlation) for the simulation of each AVHRR channel and each channel combination, and for every tested simulation method. Individual best scores are highlighted in bold numbers.**

| Channel or feature | Method | median | MAE | IQR | bias | RMS | Slope | Offset | Correlation |
|---|---|---|---|---|---|---|---|---|---|
| 0.6 μm (%) | SBA-NN | **-0.102** | **3.733** | **4.104** | -0.230 | **6.251** | 0.956 | 1.339 | **0.969** |
| | Linear-1b | 0.180 | 4.410 | 5.280 | 0.065 | 7.009 | 0.929 | 2.607 | 0.959 |
| | Linear-1a | -0.227 | 4.938 | 5.760 | **0.022** | 7.978 | 0.911 | 3.242 | 0.947 |
| | No-correction | 3.0043 | 6.514 | 7.989 | 4.038 | 9.985 | 1.067 | 1.629 | 0.947 |
| 0.9 μm (%) | SBA-NN | **-0.153** | **3.928** | **4.356** | -0.323 | **6.585** | 0.947 | 1.617 | **0.965** |
| | Linear-1b | -0.231 | 4.871 | 6.202 | -0.056 | 7.592 | 0.917 | 2.952 | 0.952 |
| | Linear-1a | -0.333 | 5.157 | 6.373 | **-0.055** | 8.176 | 0.904 | 3.430 | 0.944 |
| | No-correction | 4.176 | 7.107 | 8.580 | 4.996 | 10.578 | 1.063 | 2.721 | 0.944 |
| 3.7 μm (K) | SBA-NN | **0.035** | **1.392** | **1.523** | -0.083 | **2.362** | 0.990 | 2.698 | **0.995** |
| | Linear-1b | -0.098 | 1.771 | 2.046 | -0.132 | 2.852 | 0.976 | 6.262 | 0.993 |
| | Linear-1a | -0.419 | 2.002 | 2.665 | -0.184 | 2.953 | 0.974 | 6.735 | 0.992 |
| | No correction | 0.410 | 1.989 | 2.460 | 0.847 | 3.082 | 1.001 | 0.580 | 0.992 |
| 11 μm (K) | SBA-NN | -0.045 | **1.146** | **0.855** | -0.023 | **2.504** | 0.992 | 2.118 | 0.994 |
| | Linear-1b | **-0.028** | 1.156 | 0.902 | **-0.013** | 2.506 | 0.988 | 3.113 | 0.994 |
| | Linear-1a | -0.051 | 1.150 | 0.856 | -0.028 | 2.509 | 0.989 | 2.945 | 0.994 |
| | No correction | -0.170 | 1.172 | 0.860 | -0.148 | 2.514 | 0.988 | 2.981 | 0.994 |
| 12 μm (K) | SBA-NN | **-0.051** | **1.127** | **0.839** | -0.026 | **2.457** | 0.992 | 2.165 | 0.994 |
| | Linear-1b | -0.070 | 1.161 | 0.959 | **-0.018** | 2.469 | 0.989 | 2.934 | 0.994 |
| | Linear-1a | -0.076 | 1.155 | 0.922 | -0.023 | 2.476 | 0.990 | 2.654 | 0.994 |
| | No correction | 0.220 | 1.194 | 0.930 | 0.290 | 2.505 | 0.999 | 0.508 | 0.994 |
| 0.9 μm/0.6 μm (-) | SBA-NN | **-0.001** | **0.052** | **0.051** | **-0.001** | **0.116** | 0.894 | 0.105 | **0.951** |
| | Linear-1b | -0.018 | 0.075 | 0.089 | -0.017 | 0.138 | 1.004 | -0.021 | 0.939 |
| | Linear-1a | -0.008 | 0.068 | 0.085 | -0.003 | 0.132 | 0.966 | 0.031 | 0.940 |
| | No correction | 0.017 | 0.090 | 0.090 | 0.047 | 0.219 | 1.279 | -0.235 | 0.931 |
| 11 μm – 12 μm (K) | SBA-NN | **0.004** | **0.187** | **0.219** | 0.003 | **0.327** | 0.935 | 0.081 | **0.962** |
| | Linear-1b | 0.043 | 0.248 | 0.344 | **0.005** | 0.371 | 0.830 | 0.212 | 0.955 |
| | Linear-1a | 0.038 | 0.254 | 0.364 | -0.005 | 0.376 | 0.823 | 0.210 | 0.954 |
| | No correction | -0.340 | 0.476 | 0.530 | -0.438 | 0.631 | 0.759 | -0.146 | 0.934 |
| 11 μm – 3.7 μm (K) | SBA-NN | **0.005** | **1.174** | **1.059** | 0.060 | **2.232** | 0.946 | -0.310 | **0.968** |
| | Linear-1b | 0.063 | 1.534 | 1.420 | 0.119 | 2.765 | 1.020 | 0.254 | 0.957 |
| | Linear-1a | 0.522 | 1.811 | 2.010 | 0.156 | 2.890 | 1.069 | 0.633 | 0.959 |
| | No-correction | -0.500 | 1.837 | 1.880 | -0.995 | 3.070 | 1.092 | -0.363 | 0.961 |


**Appendix C: Specification of derived MLP networks for day, night and twilight**

**Table C1: Configuration file names for finally chosen MLP networks.**

| Configuration file | Time of day |
|---|---|
| ch7_satz_max_15_SUNZ_0_80_tdiff_120_sec_20241204.yaml | DAY |
| ch7_satz_max_15_SUNZ_80_89_tdiff_120_sec_20241204.yaml | TWILIGHT |
| ch4_satz_max_15_SUNZ_90_180_tdiff_120_sec_20241204.yaml | NIGHT |

**Code availability**

The code for the network training and the resulting networks can be found on Github (https://github.com/foua-pps/sbafs_ann).

**Author contributions**

KGK, SE, NH and EW contributed to the planning and design of the study. NH and SE developed the NN code and performed the simulations. KGK, NH, and EW carried out validation experiments and inter-comparisons. RS contributed to the general
SBAF experimental setup. All authors contributed to the analysis of the results. KGK prepared the manuscript with contributions from all co-authors.

**Competing interest**

The authors declare that they have no conflict of interest.

**Acknowledgements**

The authors would like to thank Kenneth Knapp at NOAA/STAR for providing the reduced resolution VIIRS dataset in the VGAC format. We are also grateful to Jan Fokke Meirink at the Royal Netherlands Meteorological Institute (KNMI) for fruitful discussions on the impact of simulated radiances on cloud microphysical products. The CALIOP V4.20 data were obtained from the NASA Langley Research Center Atmospheric Science Data Center at
https://asdc.larc.nasa.gov/project/CALIPSO. Finally, we thank two reviewers for very constructive comments and recommendations on the manuscript.

**Financial support**

This work was performed within the EUMETSAT CM SAF and EUMETSAT NWC SAF project frameworks and we
acknowledge the financial support of the EUMETSAT member states.

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
