# Peer review of "Extension of AVHRR-based climate data records: Exploring ways to simulate AVHRR radiances from Suomi-NPP VIIRS data"

_EGUsphere, 2025_

## Referee Comment (RC1)

**Review of egusphere-2025-379**

**Extension of AVHRR-based climate data records: Exploring ways to simulate AVHRR radiances from Suomi-NPP VIIRS data**

By: Karl-Göran Karlsson, Nina Håkansson, Salomon Eliasson, Erwin Wolters, Ronald Scheirer

**Overall Recommendation**

This paper presents a novel method to simulate AVHRR radiances of the NOAA satellites from the Suomi-NPP (VIIRS) imager. The validity of the simulation method is confirmed by comparing the cloud products based on simulated data against CALIPSO cloud products as well as against CLARA cloud products based on original AVHRR radiances.

The topic of this paper is of great interest. The proposed methods are key to developing data records for climate applications. The simulation method confirms that simulated radiance can be used to increase the frequency of available observations per day, as well as to extend the data record in time by using observations from future (or may be even past) satellite imagers.

The manuscript needs to explain better how the presented work is related to other initiatives with respect to developing homogenised records of satellite radiances from observations of multiple satellites. Moreover, the manuscript appears to have been written in haste and therefore needs to be checked and revised carefully. Among others, the authors need to review the wording of their manuscript and assure to use similar terminology throughout the manuscript. Similarly, the style of figures and tables may be harmonized better for the sake of improving the presentation of the work.

The manuscript needs some major, but mainly minor revisions before it can be published. Below some general remarks followed by a chronological list of minor points of criticisms are given.

**General Criticisms**

*Abstract*
The story line may be presented clearer. For non-insiders the objective of the paper is not clear enough. Could you have another look at it and try to introduce better the purpose of the paper.

*Introduction*
The author focuses on using the method to simulate AVHRR radiance on extending their data records with observations of future satellite, thus extending the number of years covered. However, the method can also be applied to increase the number of satellites observations per day, for example by adding simulated AVHRR radiances from MODIS data, can the authors elaborate on the possibilities of that?

*Handling changes of the SBAFs over time*

The authors derive their SBAFs using collocations between VIIRS and AVHRR during the years 2012 and 2013, offers just a brief or limited view of a situation, providing a limited rather than a comprehensive or long-term perspective. Instrument spectral response functions, like those from VIIRS, are subject to changes over time. Can the authors add a discussion explaining how to handle changes in the SBAFs over time? Note, within the FIDUCEO project much emphasis was given to quantifying channel degradation and reconstructing changes in channel spectral response over time (Rüthrich et al., 2019, https://doi.org/10.3390/rs11101165).

Using collocations of two instruments to derive SBAF is not only mitigating differences in SRF but also considering any radiometric biases these measurements may have. And these biases may not be static in time, therefore the relationship derived between the instruments may not hold for another period outside of the training. Using collocations, you are basically inter-calibrating the measurements, not just deriving spectral band adjustment factors.

If one needs to account only for the SRF differences, it is advisable to simulate both AVHRR and VIIRS measurements from hyperspectral measurements such as IASI (for IR channels) and SCIAMACHY (VIS channels) and use either of the methods suggested in the paper to derive SBAF, as you already mentioned in the manuscript.

*Link to GSICS and FIDUCEO*
The work presented is related to work done by the Global Space-based Inter-Calibration System (GSICS) international partnership, an initiative launched in 2005 by the World Meteorological Organisation (WMO) and the Coordination Group for Meteorological Satellites (CGMS). Further the work has many elements touched upon in the framework of the FIDUCEO (Fidelity and Uncertainty in Climate Data Records from Earth Observations) Horizon 2020 project funded by the European Union. Within GSICS and FIDUCEO good practices and common terminologies for doing harmonization and homogenization - all satellites are forced to look like a "reference sensor", AVHRR on NOAA-19 in this paper - of level-1 observations. Hereto:

- Can the authors explain in their introduction how the presented work relates to what is being done within GSICS and FIDUCEO?
- Can the authors use the terminology suggested in GSICS and FIDUCEO, and avoid confusing terminology in different fora?

For illustrative purposes here a schematic representation of the homogenization principles

[Figure]

*Table styles*

The paper comprises many tables. The tables all have different styles (width, font size, font type, border styles, shading styles): Although I understand it is difficult to completely align table styles, I still suggest trying to homogenize the styles of the tables in the manuscript as much as possible, following the style suggested by egusphere.

*Figure style and quality*

The paper comprises many figures. The figures different in font type, font size, resolution, eg font size of Figure 5 is much larger than of Figure 9. Could the authors try to align look and feel of the figures, to make the paper optically more attractive. As above, please try to follow style suggested by egusphere.

**Minor Criticisms**

**Introduction**

*Line 45 "Reanalysis datasets are undoubtedly capable of providing the best possible description of the Earth's atmospheric and surface state evolution, at least over the last 3-5 decades*

Also good to mention here that reanalysis is designed to be physically consistent. This is an advantage and a disadvantage at the same time, as this consistence may mast out actual information that is not well covered by the current physical description.

Please see (Roebeling et al., 2025, BAMS, accepted) who write:
*Note that reanalysis products from ECMWF, like ERA5, are indeed designed to be internally consistent but are not independent, and thus one needs to be cautious when using reanalysis data for studies asking for multiple data records that are independent of each other*

**Methodology**

*Line 75 "..estimated that SBAFs can explain up to 80 % of the variance ..."*
The number of 20% of variance not explained surprises me. It is a rather large number, that I was not expecting for narrow-band visible and near-infrared imagers. Could you give examples of the channels of instrument pairs were 20% of the variance is not

explained. In addition, I would expect that the bulk of the channels are clean channels (in which both instruments little absorption lines) and the explained variance is much higher. Please elaborate.

*Line 81 "These can be derived from direct inter-comparisons of spatio-temporally collocated measurements from the two sensors"*

A good reference for this sentence would be Meirink et al. (2013) https://doi.org/10.5194/amt-6-2495-2013

*Line 90 "Notice that our target is the third version of this sensor (AVHRR/3) as carried by the NOAA-19 satellite"*
Consider referring to the AVHRR/3 sensor on NOAA-19 as "the reference sensor"

*Line 95 "In this study, we are not interested in simulating AVHRR channel 3A, as shown in Table 1. The reason 95 is that satellites carrying the VIIRS sensor follow an afternoon orbit, a sun-synchronous path with a daytime equator crossing shortly after noon."*

Could this still become of interest in future work? This would for example be the case a future version of CLARA would try to increase the diurnal observation frequency and add simulated observation of the 'reference sensor' based on MODIS.

*Line 129 "The CLARA-A3 CDR is based on AVHRR data with a coarser resolution than the nominal horizontal resolution. The archived global AVHRR dataset is stored in a format called Global Area Coverage (GAC) with a horizontal resolution of approximately 4 km (Kidwell, 1991)."*

I suggest shortening this sentence to
"The CLARA-A3 CDR is based the archived global AVHRR dataset is stored in a format called Global Area Coverage (GAC) with a horizontal resolution of approximately 4 km (Kidwell, 1991)."

*Line 129 "..an equivalent format."*

I suggest rephrasing this part to
"..an equivalent horizontal resolution."

*Line 139 "..various SBAF relations, ..."*

Do I understand well that NASA developed a set of different SBAFs? What is the reason for having different SBAFs, is it related to the surface underneath, or atmospheric absorption features? Please elaborate.

*Line 160: Figure 2*

For clarity, could you also show the local crossing time of Suomi-NPP in this figure?

*Line 180: "The linear SBAF regression methods utilized"*

From this line I understand that the paper evaluates two SBAF methods. Could you use, throughout the paper, the same terminology for these two methods, as you do nicely in the next section, but did not earlier in the paper, eg.

1. SBAFs derived from linear regression (referred to as SBAF-Linear1a and SBAF Linear1b)
2. SBAFs derived from a MultiLayer Perceptron (MLP) neural network (referred to as SBAF-NN)

*Line 209: " eg for harmonizing"*

Do you mean here harmonization of homogenizing. The difference between the two terms are explained in the FIDUCEO project, see https://www.fiduceo.eu/vocabulary

**Harmonisation**
A harmonised satellite series is one where all the calibrations of the sensors have been made consistent with (a) reference dataset(s) which can be traced back to known reference sources, in an ideal case back to SI. Each sensor is calibrated to the reference in a way that maintains the characteristics of that individual sensor such that the calibration radiances represent the unique nature of each sensor. This means that two sensors which have been harmonised may see different signals when looking at the same location at the same time where the difference is related to known differences in the responses of each sensor such as differences in the sensors spectral response functions. Harmonisation can be achieved to within an uncertainty that should be estimated, and the uncertainty contributes to the component of uncertainty that is common across the whole record of a given sensor.

**Homogenisation**
Unlike harmonisation, homogenisation is where all satellites are forced to look the same such that when looking at the same location at the same time they would (in theory) give the same signal. In reality the signals from different sensors would be different and homogenisation is adding in corrective terms to each satellite to make them look the same. It is likely that these corrective terms will not be 100% effective and that the process of homogenisation will add in scene dependent errors to the uncertainty budget which may be difficult to assess.

*Line 210: Difference between Linear-1a and Linear-1b*
Could you provide more information about the difference between Linear-1a and Linear-1b? I ask because in Tables B1, B2, B3, and B4 both methods appear. The authors indicate that Linear-1b separates day, night, and twilight, does this mean you separated your data pairs in three groups to do the Linear-1b regression, if so, what criteria were used for separating the data pairs?

*Line 226: "Definition and training of the MLP network "*

To be consistent in terminology consider using

"Definition and training of the SBAF-NN"

*Line 300: "Table 2 … "*

Should this not be another table? Table 2 shows the spectral bands of VIIRS much earlier in the paper, but no results. Please check cross references to tables and figures throughout the manuscript.

*Tables 3, 4, 5, 6*
These are very technical tables; I propose to move these to an Appendix.

*Figure 3, 4, 5*
The plots in these figures do not show regression statistics (slope, offset, and correlation), making that I can only compare the results qualitatively. The authors provide some of these statistics in the Tables B1, B2, B3, and B4. What are the considerations to leave slope, offset, and correlation out of these tables.

Note that much of the differences in the scatterplot result from collocation and synchronisation differences and do not talk so much about the performance of the SBAF correction. In that sense Figure 6 is a much better figure to illustrate the difference between the methods. Basically, it suffices to only present Figure 6, and leave out Figure 3, 4, 5. Can the authors comment on this?

Tables B1, B2, B3, and B4 present the most important quantitative results of the paper. I propose to either make these tables part of the main text, or to refer to Appendix B in the captions of Figure 3,4,5, or make the Tables B1, B2, B3, B4 in Appendix B part of the main text?

*Figure 6*
This is a very informative figure. Similar as above, also here good to refer to table that presents the statistics to this figure, i.e. B1, B2, B3, or B4. Do the results shown in Figure 6 represent the results of for the radiance validation scores for ALL cases (Table B4)?

*Table 7, 8, 9*
The result of CFC bias and CTH bias do not match very well. The authors do not describe this in the text. I understand that the Calipso observation do only become representative when a small COT threshold is set, to ascertain that low concentration of small particles in thin air is excluded. If this is done, the results become more robust. Can to authors explain this in the text and only keep in the statistics for CFC bias (COT > 0.2) CTH bias (COT > 0.4).

*Table 7, 8, 9*
Why are the COT criteria for CFC bias (COT > 0.2) different from the one for CTH bias (COT > 0.4)?

*Line 414 "The SBAF-NN method has the best overall scores for the VIIRS/VGAC simulations validated against all CALIPSO-detected clouds."*
Judging from the statistics in Table 7 none of the methods seems to stick out with better results. It appears even that VGAC no SBAF has very good results. Please explain.

*Figure 7, 8, 9*
These figures present scatterplots of data pairs of cloud property retrieval from two different satellites. Although the authors did everything to collocate and synchronise the data pairs, much of the observed scatter is simply the result so collocation and synchronisation differences and does not talk so much about the performance of the SBAF correction. To mitigate this, it would be better to present the results are frequency distributions of differences as is done in Figure 6. Can the authors comment why they have chosen to use scatterplot instead of frequency distributions to present the results?

*Table A1 - A4*
Table A1 - A4 presents the slope and offset of the linear regression on the training sets for Linear-1a and Linear-1b. The tables mention channel numbers and not channel wavelength. As the manuscript mostly refers to the wavelength and not the number of the channel, can the authors add the wavelength in brackets for each channel?

---

## Author Comment (AC1)

**Review of egusphere-2025-379**
**Extension of AVHRR-based climate data records: Exploring ways to simulate AVHRR radiances from Suomi-NPP VIIRS data**

By: Karl-Göran Karlsson, Nina Håkansson, Salomon Eliasson, Erwin Wolters, Ronald Scheirer

**Reviewer 1**

**Overall Recommendation**

This paper presents a novel method to simulate AVHRR radiances of the NOAA satellites from the Suomi-NPP (VIIRS) imager. The validity of the simulation method is confirmed by comparing the cloud products based on simulated data against CALIPSO cloud products as well as against CLARA cloud products based on original AVHRR radiances.

The topic of this paper is of great interest. The proposed methods are key to developing data records for climate applications. The simulation method confirms that simulated radiance can be used to increase the frequency of available observations per day, as well as to extend the data record in time by using observations from future (or may be even past) satellite imagers.

The manuscript needs to explain better how the presented work is related to other initiatives with respect to developing homogenised records of satellite radiances from observations of multiple satellites. Moreover, the manuscript appears to have been written in haste and therefore needs to be checked and revised carefully. Among others, the authors need to review the wording of their manuscript and assure to use similar terminology throughout the manuscript. Similarly, the style of figures and tables may be harmonized better for the sake of improving the presentation of the work.
The manuscript needs some major, but mainly minor revisions before it can be published. Below some general remarks followed by a chronological list of minor points of criticisms are given.

Author response: *We thank the reviewer for the appreciating words and for very relevant and constructive comments. We provide our answers and explanations below.*

**General Criticisms**

*Abstract*
The story line may be presented clearer. For non-insiders the objective of the paper is not clear enough. Could you have another look at it and try to introduce better the purpose of the paper.

Author response: *Yes, we can do this.*
*We suggest the following introducing sentences in the abstract:*

"The long series of multispectral measurements from the Advanced Very High Resolution Radiometer (AVHRR), which began in 1979, is now approaching its end, with the last remaining AVHRR sensor currently operating aboard EUMETSAT's Metop-C satellite. Several Climate Data Records (CDRs) built on AVHRR data now face the end of their observational record. However, since many modern imagers contain AVHRR-heritage spectral channels, a potential for extension of these AVHRR-based climate data records exists. This study investigates the possibility to simulate original National Oceanic and Atmospheric Administration (NOAA)-19 AVHRR channels from the Suomi National Polar-orbiting Partnership (NPP) Visible Infrared Imaging Radiometer Suite (VIIRS) radiances using collocated AVHRR/VIIRS
datasets from 2012-2013. Spectral Band Adjustments…"

*Introduction*

The author focuses on using the method to simulate AVHRR radiance on extending their data records with observations of future satellite, thus extending the number of years covered. However, the method can also be applied to increase the number of satellites observations per day, for example by adding simulated AVHRR radiances from MODIS data, can the authors elaborate on the possibilities of that?

*Author response: Yes, it would certainly be possible to do a similar operation with MODIS data. In fact, we could repeat collocations with MODIS for the same years (2012-2013), since at that time all three satellites (NOAA-19, Suomi-NPP, and Aqua) had almost identical orbits. However, having already access to VIIRS data, we would not gain so much for Climate Data Records (CDRs) when adding MODIS-based products, because Suomi-NPP and NOAA-20 have nearly identical orbits compared with Aqua. Thus, Aqua-MODIS-based data would rather duplicate VIIRS-simulated AVHRR data, thus not further complementing or improving any AVHRR-based CDR. Instead, it appears more fruitful to stick with VIIRS-based data, since this would give the possibility to extend the data record many years into the future (considering that the end of MODIS observations is already at hand).*

*Hypothetically, one could perhaps think of using MODIS-simulated AVHRR data for the years 1999-2011, but this would then again mostly duplicate already existing AVHRR datasets from NOAA-16 and NOAA-18, which have almost similar afternoon orbits as Aqua, although data duplication is not the same as for the period after 2012, since both NOAA-16 and NOAA-18 in their afternoon orbits also suffered from similar substantial orbital drift as NOAA-19. So, MODIS-based results could be useful for at least 1999-2001 (see Figure 2 in the manuscript). We will consider if it is worth the effort to add MODIS-derived results, but for the moment we consider the data loss after 2012 as the most important gap to be filled. Finally, there is a worst-case scenario to consider too: Imagine we lose both Metop satellites soon and the access to METimage data becomes significantly delayed. This could lead to a gap in the access to morning orbit data. In this particular case one could consider filling this gap with products derived from Sentinel 3 Sea and Land Surface Temperature Radiometer (SLSTR) observations. There should be enough cases with overlapping Metop and SLSTR data (even if orbits are not very close), where the current methods could be used to simulate AVHRR from SLSTR. But let's hope that this scenario will never happen.*

*We suggest to add the following statement after line 597.*

"Finally, the possibility to also use MODIS-based AVHRR data simulations can be considered to improve observational coverage for some earlier years in the CLARA CDR (e.g., for 1999-2001 when the NOAA-16 orbital drift was considerable). However, highest priority is to secure a CDR extension with VIIRS and METimage data."

*Handling changes of the SBAFs over time*

The authors derive their SBAFs using collocations between VIIRS and AVHRR during the years 2012 and 2013, offers just a brief or limited view of a situation, providing a limited rather than a comprehensive or long-term perspective. Instrument spectral response functions, like those from VIIRS, are subject to changes over time. Can the authors add a discussion explaining how to handle changes in the SBAFs over time? Note, within the FIDUCEO project much emphasis was given to quantifying channel degradation and reconstructing changes in channel spectral response over time (Rüthrich et al., 2019, https://doi.org/10.3390/rs11101165).

Using collocations of two instruments to derive SBAF is not only mitigating differences in SRF but also considering any radiometric biases these measurements may have. And these biases may not be static in time, therefore the relationship derived between the instruments may not hold for another period outside of the training. Using collocations, you are basically inter-calibrating the measurements, not just deriving spectral band adjustment factors.

If one needs to account only for the SRF differences, it is advisable to simulate both AVHRR and VIIRS measurements from hyperspectral measurements such as IASI (for IR channels) and SCIAMACHY

(VIS channels) and use either of the methods suggested in the paper to derive SBAF, as you already mentioned in the manuscript.

Author response: *This is a very important point, which we think we at least have touched upon in the paper, but it is clear that it can be discussed even further. We had the degradation problem in mind (which we clearly stated on lines 287-292) when we performed a second evaluation of resulting cloud products in 2019 (about 6 years after we derived the SBAF-relations, results given in Tables 7-9). We concluded that results from 2012-2013 were basically repeated in 2019, indicating that no major degradation effects were seen. Nevertheless, results based on NOAA-20 VIIRS sensor data deviated slightly more than results from Suomi-NPP, possibly indicating that spectral response differences between the two VIIRS sensors might have some influence. However, results were still clearly within the CLARA CDR product requirements.*

*It is not possible to repeat the study based on direct AVHRR/VIIRS collocations from 2012-2013, since this time window is the only one allowing a global comparison of results. After 2013, collocations are only possible at very high latitudes (near +/- 70º), leading to poor coverage of the high temperature range. Thus, direct collocations cannot be efficiently used to monitor possible degradations over time. However, indirect investigations on the effect of derived cloud products (like the ones for 2019) are certainly possible, e.g., until 2022 with CALIPSO as reference and hopefully after 2025 with EarthCARE as reference. These indirect investigations can certainly be planned with regular intervals to at least get a rough indication of the VIIRS spectral responses' stability and the applied SBAF relations.*

*More problematic is probably the use of hyperspectral measurements. We mentioned already in section 2.4 that we first tested the use of hyperspectral methods to derive SBAFs based on the datasets prepared by NASA. In fact, it was initially assumed that this would be accurate enough. But this did not work out well. The SBAFs from the NASA datasets did not give us the needed accuracy to be able to reproduce AVHRR radiances from VIIRS. Especially the 3.7 micron channel failed here, which led to substantial problems for cloud detection at night (when data from this channel are crucial). The fact that the IASI spectral coverage does not include the entire spectral range of the 3.7 micron channel (i.e., the shortest wavelengths in this channel's spectral response are not covered by IASI) is probably one of the reasons for this failure. In addition, the coarser horizontal resolution of IASI data (25 km) could also lead to some differences. A third alternative could be that the geographic sampling of IASI measurements was too limited. Because of these factors, we decided to better trust direct AVHRR/VIIRS collocations and this motivation is clearly described in the manuscript. We must also claim that the study of collocated data with approximately the same horizontal resolution facilitated the derivation of maybe the study's most interesting result, namely the ability to achieve object-dependent SBAFs for the 3.7micron AVHRR channel (as demonstrated in Figure 11 and Table 10). This would have been difficult to achieve with the coarser resolution IASI measurements. Notice that this also required access to visible information for being successful.*

*The idea to try using IASI (or other) hyperspectral measurements for evaluation of the spectral responses' stability requires that real IASI/VIIRS collocations can also be made on a regular basis. This is problematic, since IASI is carried on polar satellites in a mid-morning orbit (Metop satellites), while VIIRS is in an afternoon orbit. Again, this means that collocations can only be made at very high latitudes. This, in turn, means that evaluation of VIIRS radiances can only be made for relatively low target temperatures. Another option could be to instead use measurements from the CrIS sensor carried together with VIIRS on the Suomi-NPP and NOAA satellites. This would give continuous matchups globally. But what is very unfortunate here is that CrIS has an even more restricted coverage of the 3.7 micron AVHRR channel, since measurements only start at 3.9 microns and thus cover the lower end of the 3.7 micron spectral response even worse than IASI. This limitation is quite serious with respect to the influence of the solar reflected part in the 3.7 micron channel, occurring in particular at the shortest covered wavelengths of this channel.*

*Investigations of the stability of the visible VIIRS channels could be tested using the TROPOMI sensor on Sentinel 5p. Here, global collocations with VIIRS are clearly possible, since orbit characteristics are*

*very similar. Unfortunately, the spectral ranges of this sensor do not cover the typical wavelengths for the AVHRR or AVHRR heritage channels (e.g. at 0.6 and 1.6 microns), so the opportunity to use TROPOMI are unfortunately not there at all. The best alternative is probably to check visible channels from other hyperspectral measurements. One possibility would be the Ocean Color Instrument (OCI) sensor on NASA's Plankton, Aerosol, Cloud, ocean Ecosystem (PACE) mission or the JPL Earth surface Mineral dust source Investigation (EMIT) mission [https://www.spiedigitallibrary.org/conference-proceedings-of-spie/13267/132670B/Inter-comparison-of-NOAA-21-NOAA-20-S-NPP-VIIRS/10.1117/12.3040408.short](https://www.spiedigitallibrary.org/conference-proceedings-of-spie/13267/132670B/Inter-comparison-of-NOAA-21-NOAA-20-S-NPP-VIIRS/10.1117/12.3040408.short)). Also, comparisons could be done with Sentinel 3 Ocean and Land Colour Instrument (OLCI) measurements. Although not purely hyperspectral, OLCI contains 21 bands in the 0.4-1.0 micron range and has an overpass time of 10:00 AM.*

*In conclusion, the prospects for performing a continuous monitoring of VIIRS spectral responses and their stability by using hyperspectral measurements are unfortunately very limited, since the AVHRR channels are either not or only partially covered by sensors available for the nearest future. On a longer term this can change (e.g. through introduction of missions like Climate Absolute Radiance and Refractivity Observatory (CLARREO)), but this will take some time. We conclude that collocations with IASI and a continuous check of derived cloud products from the simulated radiances based on EarthCARE or other future reference data are currently the best ways to monitor the stability of measurements until more reliable and more accurate reference measurements are available.*

*Based on the above discussion, we propose the following additions to the text (from line 591 onwards):*

"In order to check the validity of the derived spectral adjustments on a longer timescale, it is suggested to regularly repeat the validation efforts on derived cloud products based on new observations from active sensors on EarthCARE (Illingworth et al., 2015) and similar missions in the future. In addition, regular checks of Simultaneous Nadir Observations (SNOs) at high latitudes between VIIRS and IASI (and its successors) can also help in deducing the infrared bands spectral responses' stability. The possibility to make similar checks for VIIRS and METimage visible bands seems unfortunately not possible from Troposphere Ozone Monitoring Instrument (TROPOMI, Veefkind et al., 2012) due to very limited spectral coverage of the AVHRR-heritage channels. However, it is possible that measurements from the Ocean Color Instrument (OCI) sensor on the Plankton, Aerosol, Cloud, and ocean Ecosystem (PACE, Gorman et al., 2019) mission and the Earth surface Mineral dust source Investigation (EMIT, Connely et al., 2021) sensor onboard the International Space Station (ISS) can be used. Some reference measurements from the Sentinel 3 Ocean and Land Colour Instrument (OLCI, Donlon et al., 2012) measurements could also be used. Although not purely hyperspectral, OLCI contains 21 bands in the 0.4-1.0 µm range and has an equator overpass time at 10:00 AM."

*Link to GSICS and FIDUCEO*
The work presented is related to work done by the Global Space-based Inter-Calibration System (GSICS) international partnership, an initiative launched in 2005 by the World Meteorological Organisation (WMO) and the Coordination Group for Meteorological Satellites (CGMS). Further the work has many elements touched upon in the framework of the FIDUCEO (Fidelity and Uncertainty in Climate Data Records from Earth Observations) Horizon 2020 project funded by the European Union. Within GSICS and FIDUCEO good practices and common terminologies for doing harmonization and homogenization - all satellites are forced to look like a "reference sensor", AVHRR on NOAA-19 in this paper - of level-1 observations. Hereto:

• Can the authors explain in their introduction how the presented work relates to what is being done within GSICS and FIDUCEO?

• Can the authors use the terminology suggested in GSICS and FIDUCEO, and avoid confusing terminology in different fora?

For illustrative purposes here a schematic representation of the homogenization principle:

[Figure]

Author response: *We certainly should have used the appropriate terminology introduced by GSICS and FIDUCEO and we have therefore now corrected this by completely rewriting Section 2.1 and also adjusted some further paragraphs in subsequent sections. We found it best to introduce these definitions here and not in the Introduction part. So, we propose a completely new introduction of Section 2.1:*

"The problem to adjust measurements after introduction of a slightly modified or new version of a sensor is an old problem and it has been subject to substantial efforts over the years. Most well-known are the activities of the Global Space-based Inter-Calibration System (GSICS, WMO 2025), in which the primary goal is to ensure a homogeneous behavior of measurement time series from a particular sensor or spectral channel. We call this adjustment "Inter-calibration" (Chander et al., 2013a) and the purpose here is to provide a homogenous data record without artificial discontinuities. The Fidelity and uncertainty in climate data records from Earth Observations project (FIDUCEO) emphasized the difference between homogenized and harmonized data sets (Giering et al., 2019) with relevance for the CDR compilation. Harmonized data would imply corrections to a measurement based on high-quality reference measurements, thus providing the best possible estimation of the measured radiance. This correction would still allow differences to a similar instrument having slightly different spectral responses. However, for a CDR, which should allow for climate trends estimation, homogenized data seemingly should be the best choice. On the other hand, it could also lead to sensor accuracy violation (if the various sensors have significant differences in spectral response). Thus, there are pros and cons of both spectral adjustment methods and any of them shall be applied with caution. An important aspect is also that radiance differences between two sensors might be caused by additional factors other than differences in spectral responses, e.g. radiance biases or calibration errors.

When focusing on the current problem to simulate AVHRR from VIIRS radiances based solely on spectral response differences, no SBAF spectral adjustment methodology will ever be able to simulate AVHRR channels perfectly, since some parts of the spectrum covered by another AVHRR-heritage sensor channel are simply not observed by the corresponding AVHRR channel (and vice versa)…"

*However, we also want to highlight the fact that we have some different data treatment from visible and infrared AVHRR channels. Below follows a short description of the data used in the CLARA CDR.*

*In the CLARA CDR, we have so far used an FCDR containing harmonized AVHRR data based on a method developed by Heidinger (2018) for the visible channels. This FCDR is based on several input data consisting of MODIS/AVHRR SNOs, AVHRR/AVHRR SNOs and comparisons with stable terrestrial sites (Libyan desert, Greenland, Dome C, etc.). Measurements are compared over several years and an average reflectance is calculated including also a temporal correction term to account for sensor degradation (see example figure below for AVHRR channel 1 from NOAA-19 showing calibration curves from 2017 as used in CLARA-A3). Thus, here we used harmonized data (unique for every AVHRR sensor and visible channel), i.e., no intercalibration, thus NOT homogenized - and with a temporal correction. However, there is indeed a small element of inter-calibration between AVHRRs in that also AVHRR-to-AVHRR SNOs (marked as "virtual" in the figure below) are involved in the process. The adjustment between MODIS and AVHRR spectral channels is based on SBAFs derived by NASA (the same dataset as we have already tested and referred to in the manuscript).*

[Figure]

*For infrared data we still have no access to a true FCDR (FIDUCEO only produced av very preliminary version), which means that we have relied on the onboard operational blackbody calibration procedure from NOAA. For the next CLARA major edition (CLARA-A4) we anticipate to use a full AVHRR FCDR for all AVHRR channels provided by the EUMETSAT Secretariat.*

*In conclusion, for the simulated VIIRS-based AVHRR dataset, the reference is harmonized data from NOAA-19. We do not see a problem in not having used homogenized data here, since the intention from the start has been to replace the loss of NOAA-19 observations in the afternoon orbit. In practice, it means that the reference for the AVHRR simulations from VIIRS is NOAA-19, which in turn means that all future AVHRR simulations from VIIRS will also be of type NOAA-19 (but certainly subject to changes depending on how individual VIIRS sensors perform, as has been discussed earlier).*
*Finally, we will clarify the status of the NOAA-19 AVHRR data on line 94:*

"It should also be mentioned that the reference radiances for NOAA-19 AVHRR should be considered as harmonized data, since their quality and evolution over time has been optimized for this particular AVHRR sensor by a method described by Heidinger (2018)."

*Table styles*
The paper comprises many tables. The tables all have different styles (width, font size, font type, border styles, shading styles): Although I understand it is difficult to completely align table styles, I still suggest trying to homogenize the styles of the tables in the manuscript as much as possible, following the style suggested by egusphere.

Author response: *Yes, we admit that table styles are not consistent in the manuscript. We will improve and use the same table style for all tables in the manuscript.*

*Figure style and quality*
The paper comprises many figures. The figures different in font type, font size, resolution, eg font size of Figure 5 is much larger than of Figure 9. Could the authors try to align look and feel of the figures, to make the paper optically more attractive. As above, please try to follow style suggested by egusphere.

Author response: *Yes, we agree that figures should have been better harmonized. However, we have now decided to skip scatterplots in Figures 7-10 (upon your suggestion) and instead only present difference/frequency plots. See comment further down about this.*

**Minor Criticisms**

**Introduction**
*Line 45 "Reanalysis datasets are undoubtedly capable of providing the best possible description of the Earth's atmospheric and surface state evolution, at least over the last 3-5 decades*
Also good to mention here that reanalysis is designed to be physically consistent. This is an advantage and a disadvantage at the same time, as this consistence may mast out actual information that is not well covered by the current physical description.
Please see (Roebeling et al., 2025, BAMS, accepted) who write:
*Note that reanalysis products from ECMWF, like ERA5, are indeed designed to be internally consistent but are not independent, and thus one needs to be cautious when using reanalysis data for studies asking for multiple data records that are independent of each other*
Author response: *We will add the following comment on line 44:*

"..a multitude of global observations and the use of a physically consistent methodology based on model physics constraints."

*We also propose to refer to your statement in the BAMS paper on line 49:*

"Furthermore, the reanalysis dependency on physical constraints from the current Numerical Weather Prediction (NWP) model means that the results are not completely independent, since model physics cannot be considered as perfectly describing the real atmosphere/Earth system (as pointed out by Roebeling et al., 2025)."

**Methodology**
*Line 75 "..estimated that SBAFs can explain up to 80 % of the variance ..."*
The number of 20% of variance not explained surprises me. It is a rather large number, that I was not expecting for narrow-band visible and near-infrared imagers. Could you give examples of the channels of instrument pairs were 20% of the variance is not explained. In addition, I would expect that the bulk of the channels are clean channels (in which both instruments little absorption lines) and the explained variance is much higher. Please elaborate.

Author response: *This statement is not ours, but taken from the study of Piontek et al., 2023. But it seems we phrased it slightly wrongly. They write "The spectral band adjustment factors can remove the bias and even reduce the standard deviation in the brightness temperature difference by **more than 80%.**" Thus, 80 % is not the maximum amount, but even higher values of the explained variance can be seen. So, I guess it depends on how closely overlapping the channels are. If they are very closely overlapping the potential is, of course, higher to get a good agreement. We will modify our statement slightly on line 76 and we regret our previous formulation:*

"Piontek et al. (2023) estimated that linear SBAFs can explain more than 80 % of the variance, but the efficiency depends on the selected channels."

*Line 81 "These can be derived from direct inter-comparisons of spatio-temporally collocated measurements from the two sensors"*
A good reference for this sentence would be Meirink et al. (2013) https://doi.org/10.5194/amt-6-2495-2013

Author response: *Yes, we will add that reference on line 81.*

Line 90 *"Notice that our target is the third version of this sensor (AVHRR/3)
as carried by the NOAA-19 satellite"*
Consider referring to the AVHRR/3 sensor on NOAA-19 as "the reference sensor"

Author response: *OK, we will adopt that notation on line 90.*

Line 95 *"In this study, we are not interested in simulating AVHRR channel 3A, as shown in Table 1. The
reason is that satellites carrying the VIIRS sensor follow an afternoon orbit, a sun-synchronous path
with a daytime equator crossing shortly after noon."*
Could this still become of interest in future work? This would for example be the case a future version
of CLARA would try to increase the diurnal observation frequency and add simulated observation of
the 'reference sensor' based on MODIS.

Author response: *Certainly, but the main thing here is that we do not have access to measurements from
this channel, as NOAA-19 is not operating this channel for the chosen collocation period. But the
MODIS example is not very relevant (which has been explained earlier), since adding MODIS-derived
products do not necessarily increase the diurnal observational coverage. Another problem is that we
want to have consistency for all satellites operating in an afternoon orbit in the CLARA CDR. To
introduce simulated observations from channel 3A in the series of afternoon observations would break
the data record's consistency (e.g., all earlier NOAA satellites in afternoon orbit use the 3B channel).
We prefer to still limit the use of channel 3A observations to satellites in morning orbits. It is more
relevant to use both channels 3A and 3B for sensors where both channels operate simultaneously (e.g.
on modern geostationary sensors).*

Line 129 *"The CLARA-A3 CDR is based on AVHRR data with a coarser resolution than the nominal
horizontal resolution. The archived global AVHRR dataset is stored in a format called Global Area
Coverage (GAC) with a horizontal resolution of approximately 4 km (Kidwell, 1991)."*
I suggest shortening this sentence to
"The CLARA-A3 CDR is based on the archived global AVHRR dataset stored in a format called Global
Area Coverage (GAC) with a horizontal resolution of approximately 4 km (Kidwell, 1991)."

Author response: *OK, to be adopted.*

Line 129 *"..an equivalent format."*
I suggest rephrasing this part to
"..an equivalent horizontal resolution."

Author response: *OK, to be adopted.*

Line 139 *"..various SBAF relations, ..."*
Do I understand well that NASA developed a set of different SBAFs? What is the reason for having
different SBAFs, is it related to the surface underneath, or atmospheric absorption features? Please
elaborate.

Author response: *Well, this comes just from the fact that the NASA SBAF tool allows several options for
how the SBAFs can be calculated from their database. Different constraints can be set on the linear
calculations (e.g., forcing relation through zero, etc.) and even non-linear relations (quadratic) can be
derived from the data. Also, one can choose to restrict calculations to be either global, regional or
restricted to certain objects or surfaces. So, the statement is just acknowledging the great flexibility of
the NASA SBAF tool.*

Line 160: Figure 2

For clarity, could you also show the local crossing time of Suomi-NPP in this figure?

Author response: *Figure 2 is a standard figure showing all the satellites used in the CLARA-A3 dataset (e.g., used in the CLARA-A3 paper in ESSD in 2013). We prefer to leave this figure unchanged and instead underline in the caption that Suomi-NPP's crossing time is stable at 13:30 LST.*

*Line 180: "The linear SBAF regression methods utilized"*
From this line I understand that the paper evaluates two SBAF methods. Could you use, throughout the paper, the same terminology for these two methods, as you do nicely in the next section, but did not earlier in the paper, eg.
1.              SBAFs derived from linear regression (referred to as SBAF-Linear1a and SBAF Linear1b)
2.              SBAFs derived from a MultiLayer Perceptron (MLP) neural network (referred to as SBAF-NN)

Author response: *Yes, we evaluated two linear SBAF methods and one method based on a neural network (so, actually three methods were studied). The latter will now be denoted as SBA-NN in the revised paper to not confuse it with standard SBAF terminology (mostly being used for linear relations). However, we actually think we have a correct and consistent description of the different methods in the manuscript. The reviewer is probably misinterpreting the two previous points (numbered 1 and 2 as listed above). Number 1 results in two different linear approaches (Linear-1a and Linear 1-b, as described on lines 213-214) and number 2 results in one method based on neural networks (denoted SBA-NN). We think we consistently follow these notations throughout the remainder of the paper.*

*Line 209: " eg for harmonizing"*
Do you mean here harmonization of homogenizing. The difference between the two terms are explained in the FIDUCEO project, see https://www.fiduceo.eu/vocabulary
**Harmonisation**
A harmonised satellite series is one where all the calibrations of the sensors have been made consistent with (a) reference dataset(s) which can be traced back to known reference sources, in an ideal case back to SI. Each sensor is calibrated to the reference in a way that maintains the characteristics of that individual sensor such that the calibration radiances represent the unique nature of each sensor. This means that two sensors which have been harmonised may see different signals when looking at the same location at the same time where the difference is related to known differences in the responses of each sensor such as differences in the sensors spectral response functions. Harmonisation can be achieved to within an uncertainty that should be estimated, and the uncertainty contributes to the component of uncertainty that is common across the whole record of a given sensor.
**Homogenisation**
Unlike harmonisation, homogenisation is where all satellites are forced to look the same such that when looking at the same location at the same time they would (in theory) give the same signal. In reality the signals from different sensors would be different and homogenisation is adding in corrective terms to each satellite to make them look the same. It is likely that these corrective terms will not be 100% effective and that the process of homogenisation will add in scene dependent errors to the uncertainty budget which may be difficult to assess.

Author response: *We acknowledge that the use of the word "harmonizing" in this sentence is not appropriate. We have addressed both harmonization and homogenization of measurements in earlier points and this will also be reflected in the updated manuscript. We emphasize again that our application here is based on harmonized data from the NOAA-19 AVHRR sensor, so homogenization is not an issue. But in this case, it is unnecessary to mention either of the terms. We updated this sentence as follows:*

"Method 1 is the classical regression method, often used in inter-calibration applications relating two nearby spectral channels, in which measurements can be collocated during an overlapping period."

*Line 210: Difference between Linear-1a and Linear-1b*

Could you provide more information about the difference between Linear-1a and Linear-1b? I ask because in Tables B1, B2, B3, and B4 both methods appear. The authors indicate that Linear-1b separates day, night, and twilight, does this mean you separated your data pairs in three groups to do the Linear-1b regression, if so, what criteria were used for separating the data pairs?

Author response: *The reviewer has interpreted it correctly: Linear-1a is based on all data, Linear-1b separated the training dataset into three categories valid for night, twilight, and day, with different regression results for each category. This is clearly stated on lines 213-214. We do not think we need to clarify this further.*
*The tables in the Appendix A (A1-A4) show the resulting regression coefficients (with A1 valid for all data in Linear-1a and A2-A4 for the three individual categories in Linear-1b). The results presented in Appendix B includes different results for all methods separated into the three categories Day, Twilight, Night plus the overall results averaging all results. The intention here is that, by separating validation results in this way, it is possible to evaluate the impact of using a finer partitioning of day categories for the Linear 1b version versus the Linear 1a version. Of interest here is, of course, also to compare with the corresponding results for the SBA-NN method for the different categories.*

Line 226: *"Definition and training of the MLP network "*
To be consistent in terminology consider using
"Definition and training of the SBAF-NN"

Author response: *OK, adopted but changed to "Definition and training of SBA-NN".*

Line 300: *"Table 2 ... "*

Should this not be another table? Table 2 shows the spectral bands of VIIRS much earlier in the paper, but no results. Please check cross references to tables and figures throughout the manuscript.

Author response: *This sentence is definitely wrong. Thanks for discovering it. It is now removed.*

Tables 3, 4, 5, 6
These are very technical tables; I propose to move these to an Appendix.

Author response: *We admit that the tables are loaded with a lot of information, which can be a bit heavy to digest. However, we consider Tables 3-5 as quite important for the understanding of how training was done and how the chosen network was defined so we prefer keeping them as they are. Table 6 is not necessary to have in the main text and it is moved to Appendix C.*

Figure 3, 4, 5
The plots in these figures do not show regression statistics (slope, offset, and correlation), making that I can only compare the results qualitatively. The authors provide some of these statistics in the Tables B1, B2, B3, and B4. What are the considerations to leave slope, offset, and correlation out of these tables.
Note that much of the differences in the scatterplot result from collocation and synchronisation differences and do not talk so much about the performance of the SBAF correction. In that sense Figure 6 is a much better figure to illustrate the difference between the methods. Basically, it suffices to only present Figure 6, and leave out Figure 3, 4, 5. Can the authors comment on this?
Tables B1, B2, B3, and B4 present the most important quantitative results of the paper. I propose to either make these tables part of the main text, or to refer to Appendix B in the captions of Figure 3,4,5, or make the Tables B1, B2, B3, B4 in Appendix B part of the main text?

Author response: *We did consider including a set of statistical parameters in the plots, but concluded that this would make figures overloaded with information. That's why we concluded that it is better to collect statistics as separate tables in Appendix B. We propose here to keep tables in Appendix B but to also add the missing regression parameters in these tables. Notice also that in Figure 6 and in the tables we will switch the order from AVHRR-VIIRS to VIIRS-AVHRR. We found it to be more logical to present the differences that way.*

*It is true that the most interesting summarizing results are given in Figure 6, but we also think that the information in Figures 3-5 provides complementary information that better show how the data distribution really looks. For example, the results for the 0.6 micron channel in the top row of Figure 3, together with the corresponding difference plot in Figure 6 (top left) give a more complete picture of the distribution than if just using the plot in Figure 6. Thus, we still believe that presentation of all figures is motivated.*

*Figure 6*
This is a very informative figure. Similar as above, also here good to refer to table that presents the statistics to this figure, i.e. B1, B2, B3, or B4. Do the results shown in Figure 6 represent the results for the radiance validation scores for ALL cases (Table B4)?

Author response: *Thank you for the appreciating words. The answer to the question is: yes.*

*Table 7, 8, 9*
The result of CFC bias and CTH bias do not match very well. The authors do not describe this in the text. I understand that the Calipso observation do only become representative when a small COT threshold is set, to ascertain that low concentration of small particles in thin air is excluded. If this is done, the results become more robust. Can to authors explain this in the text and only keep in the statistics for CFC bias (COT > 0.2) CTH bias (COT > 0.4).

Author response: *We tend to disagree with the reviewer in the sense that there is no obvious reason why CFC and CTH results should always agree or even correlate. We are sure that this problem should be well-known in your own environment working with people who deal with, for example, the validation of OCA cloud top heights.*

*The reason is that CTH validation can only be applied on clouds that are detected. If clouds are missed by the cloud screening, CTH validation becomes less reliable unless you put restrictions on what clouds to focus on (see comments below on application of COT thresholds). It also means that if CTH retrievals are made on clouds that are detected with low (but sufficient) probability (e.g. just above 50 %), the likelihood is high that these clouds are very, very thin. We also know from experience that CTH determination is always more difficult for very thin clouds, since a correction for semi-transparency is crucial to perform in these cases. For those reasons, it is not wise to try to evaluate CTH for all detected clouds. Normally, one tries to avoid dealing with clouds that are very thin. That's why we have, in addition, included results based on different optical thickness thresholds for clouds being validated for CTH and CFC, respectively, and we have commented this in the text.*

*The COT threshold of 0.2 for validation of CFC is well-documented, e.g., in the CLARA-A2 validation paper by Karlsson and Håkansson (2018). Notice the improvement from an earlier value of 0.3 as reported by Karlsson and Johansson (2013, https://amt.copernicus.org/articles/6/1271/2013/amt-6-1271-2013-relations.html) that was seen when going from CLARA-A1 to CLARA-A2. The value has improved further for CLARA-A3, but only marginally. Down to this threshold value, clouds are generally detected well by the current CMAPROB method, i.e., with 50 % or higher probability, and the use of this threshold consequently gives the best and most reliable validation scores for CFC compared to not doing any COT thresholding at all. But, we repeat, if we are able to detect thinner and thinner clouds, it doesn't automatically also lead to CTH improvements, simply because the complexity for the semi-transparency correction increases for thinner clouds.*

*The threshold to be used for CTH validation is currently more arbitrarily chosen. It should be higher than 0.2, but not drastically higher, since cloud detection efficiency increases rapidly for COTs larger than 0.2 (as illustrated by Karlsson and Håkansson, 2018, Figure 6). A value of 0.4 was found reasonable to at least avoid some of the uncertainties associated with very thin clouds. At the same time, the threshold should not be too high, so that it does not give justice to all efforts to correct for semi-transparency (i.e., it would not make sense to only look at opaque clouds). We could have produced results for a wide range of COT thresholds, but we found it appropriate to stay with the established method that we have recently used regularly for evaluation of results from CLARA-A2 and CLARA-A3 in standard EUMETSAT validation and review studies. We also don't think that results should be restricted to only include COT thresholded results, since it always has a value to see unfiltered results, as it is generally easier to see improvements here when introducing new or upgraded methods (at least for CFC).*

*In conclusion: We insist on staying with the presented results, since in depth sensitivity analyses would otherwise be necessary for a deeper understanding. This is out of scope of this study and worth a completely separate paper in our view.*

*PS. Some confusion in the CTH results is caused by the use of mean errors (instead of the more appropriate mean absolute errors and median errors) in the current CLARA-A3 requirements (which we also mention in the manuscript on lines 416-423). This is something that needs to be changed and has already been done in corresponding validation activities of CTH parameters in the NWC SAF project, in which similar CTH retrieval methods are used.*

*Table 7, 8, 9*
Why are the COT criteria for CFC bias (COT > 0.2) different from the one for CTH bias (COT > 0.4)?

Author response: *See explanation in the previous point. We have added a comment on this on line 400:*

"The use of different COT thresholds here is motivated by the wish to avoid the thinnest clouds detected by the cloud masking procedure, since these are always the most difficult clouds to deal with for the CTH retrieval."

*Line 414 "The SBAF-NN method has the best overall scores for the VIIRS/VGAC simulations validated against all CALIPSO-detected clouds."*
Judging from the statistics in Table 7 none of the methods seems to stick out with better results. It appears even that VGAC no SBAF has very good results. Please explain.

Author response: We realize that we were incomplete and inconsistent in our description. The fact is that one can consider two different versions of the "No SBAF" condition:

1. Do no corrections at all, i.e., just process the five AVHRR-heritage channels from VIIRS in the **VIIRS processing environment of the PPS software**.
2. Use the (unchanged) AVHRR-heritage channels from VIIRS in the **AVHRR processing environment of the PPS software**.

The version that previously was referred to as "No SBAF" in the manuscript was version 1 above. But, to really analyze the effect of introducing spectral band adjustments (compared to not introducing corrections at all), we should rather have compared to version 2 above, simply since all the SBAF- and SBA-NN versions were processed in the AVHRR processing environment.

PPS is a software package which enables cloud product processing for a wide range of imagers. The shift from one sensor to another is generally dealt with by adjusting pre-calculated cloud detection thresholds, atmospheric corrections and other adaptations from mainly RTTOV-simulations utilizing

each sensors' spectral response functions. Thus, results based on version 1 can be expected to be as good (and maybe even better due to improved radiometric quality) than results based on original NOAA-19 AVHRR data. This explains why results by the previous "No SBAF" solution were comparable with original AVHRR results from CLARA-A3.

Thus, we have now also added results for version 2 in the manuscript. We will now call version 1 "PPS VIIRS" and version 2 "VGAC No SBA" in the updated result tables 7-9 and in figures. Please notice the redefinition of "VGAC No SBA" compared to the "No SBAF" definition in the original manuscript. The three tables now look as follows (notice the new column 4 with results for "VGAC No SBA"):

**Table 7. Validation scores for Cloud Fractional Cover (CFC): bias [%], Kuipers score, and hitrate (both [-]). See text and Sect. 3.3 for details.**

| Parameter | NOAA-19 CLARA-A3 | PPS VIIRS | VGAC No SBA | VGAC Linear-1a | VGAC Linear-1b | VGAC SBA-NN | VGAC SNPP 2019 SBA-NN | VGAC NOAA-20 2019 SBA-NN | CLARA-A3 Product require-ment |
|---|---|---|---|---|---|---|---|---|---|
| Total # Orbits | 1026 | 497 | 497 | 497 | 497 | 497 | 289 | 274 | |
| Total # FOVs | 6 355 780 | 3 468 354 | 3 468 354 | 3 468 354 | 3 468 354 | 3 468 351 | 1 891 597 | 1 804 134 | |
| **CFC** bias | -11.03 % | -10.46 % | -7.47 | -6.11 % | -10.20 % | -9.74 % | -10.37 % | -9.43 % | - |
| **CFC** bias (COT > 0.2) | -0.22 % | 0.28 % | 3.27 % | 4.63 % | 0.55 % | 1.00 % | 0.67 % | 1.63 % | 5 % |
| **CFC** Kuipers | 0.687 | 0.701 | 0.639 | 0.606 | 0.688 | 0.695 | 0.694 | 0.687 | - |
| **CFC** Kuipers (COT > 0.2) | 0.706 | 0.712 | 0.646 | 0.638 | 0.710 | 0.700 | 0.701 | 0.685 | 0.6 |
| **CFC** Hitrate | 0.825 | 0.833 | 0.822 | 0.814 | 0.829 | 0.834 | 0.831 | 0.833 | - |

**Table 8. Validation scores for Cloud Top Height (CTH): bias and mean absolute error (both [m]). Total number of used orbits and samples are given in Table 7. See text and Sect. 3.3 for details.**

| Parameter | NOAA-19 CLARA-A3 | PPS VIIRS | VGAC No SBA | VGAC Linear-1a | VGAC Linear-1b | VGAC SBA-NN | VGAC SNPP 2019 SBA-NN | VGAC NOAA-20 2019 SBA-NN | CLARA-A3 Product require-ment |
|---|---|---|---|---|---|---|---|---|---|
| **CTH** bias | -900 m | -1049 m | -1129 m | -504 m | -501 m | -598 m | -633 m | -408 m | - |
| **CTH** bias (COT > 0.4) | 807 m | 644 m | 641 m | 1166 m | 1143 m | 1034 m | 1122 m | 1288 m | 800 m |
| **CTH** mean abs error | 1664 m | 1678 m | 1755 m | 1560 m | 1538 m | 1541 m | 1673 m | 1656 m | - |

**Table 9. Validation scores for Cloud Phase (CPH, here the fraction of liquid clouds): bias [%], Kuipers score, and hitrate (both [-]). Total number of used orbits and samples are given in Table 7. See text and Sect. 3.3 for details.**

| Parameter | NOAA-19 CLARA-A3 | PPS VIIRS | VGAC No SBA | VGAC Linear-1a | VGAC Linear-1b | VGAC SBA-NN | VGAC SNPP 2019 SBA-NN | VGAC NOAA-20 2019 SBA-NN | CLARA-A3 Product require-ment |
|---|---|---|---|---|---|---|---|---|---|
| **CPH** mean bias | -1 % | -4 % | -6 % | -1 % | 1 % | 0 % | 0 % | 2 % | 5 % |
| **CPH** Kuipers | 0.67 | 0.66 | 0.66 | 0.69 | 0.68 | 0.68 | 0.67 | 0.68 | 0.6 |
| **CPH** Hitrate | 0.84 | 0.83 | 0.83 | 0.85 | 0.85 | 0.84 | 0.84 | 0.84 | - |

If now comparing results in columns "PPS VIIRS" and "VGAC No SBA", we notice that to process VIIRS data for AVHRR-heritage channels in the correct PPS VIIRS environment generally gives better results than when processing these data in the AVHRR PPS environment. It proves that there is indeed a need to apply spectral band adjustments to properly use VIIRS-based data in the AVHRR PPS environment. And the best method to accomplish comparable results to both original NOAA-19 AVHRR and VIIRS-only based results is to use the VGAC SBA-NN method. Notably, the "VGAC Linear-1a" method actually worsens the results for the CFC parameter. This shows the importance of separating conditions into day, night, and twilight categories.

We hope that this better demonstrates what the various spectral band adjustments are capable of. Also, it clearly demonstrates that the alternative "PPS VIIRS" (without any radiance corrections) gives equally good results compared to original NOAA-19 AVHRR data. Consequently, for the CLARA cloud products, it could have worked to use this solution as an alternative to the SBA-based solutions. However, since CLARA is an AVHRR-based CDR, one could not completely exclude (despite the good validation results) that such a solution would introduce features that are typical for VIIRS and not for AVHRR. In addition, even if the PPS software is well adapted to cope with original AVHRR-heritage VIIRS data, it is clear that many other retrieval methods used in the CLARA CDR would need to make quite extensive adaptations for allowing such a solution. For example, the clearly indicated differences between the VIIRS AVHRR-heritage visible channels M5 and M7, and corresponding AVHRR channels 1 and 2 (see Figure 3 in the manuscript) make it clear that adjustments are necessarily required at least for the surface albedo (SAL) and TOA reflected solar fluxes (RSF) products. From that perspective, it is more consistent to transfer measurements to be dealt with entirely within the AVHRR processing environment.

*Figure 7, 8, 9*
These figures present scatterplots of data pairs of cloud property retrieval from two different satellites. Although the authors did everything to collocate and synchronise the data pairs, much of the observed scatter is simply the result so collocation and synchronisation differences and does not talk so much about the performance of the SBAF correction. To mitigate this, it would be better to present the results are frequency distributions of differences as is done in Figure 6. Can the authors comment why they have chosen to use scatterplot instead of frequency distributions to present the results?

Author response: We decided to exclusively show frequency distributions in the same way as previously done for radiance comparisons in Figure6.

We give here an example for cloud effective radius (CRE) parameter:

[Figure]

With reference to the previous discussions and the introduction of the new category "No SBA", it is clear that to use VIIRS radiances without corrections in the AVHRR processing environment leads to particular problems for the CRE parameter (dotted green line in the figure). This problem seems to be solved after applying the SBA-NN corrections.

*Table A1 - A4*
Table A1 - A4 presents the slope and offset of the linear regression on the training sets for Linear-1a and Linear-1b. The tables mention channel numbers and not channel wavelength. As the manuscript mostly refers to the wavelength and not the number of the channel, can the authors add the wavelength in brackets for each channel?

Author response: *We think it should be enough to just refer to Tables 1 and 2 in the captions to not overload the table with information. Thus, we have added to the captions*

"For details on channel wavelengths for the two sensors, see Table 1 and Table 2."

---

## Author Comment (AC2)

**Review of egusphere-2025-379**
**Extension of AVHRR-based climate data records: Exploring ways to simulate AVHRR radiances from Suomi-NPP VIIRS data**

By: Karl-Göran Karlsson, Nina Håkansson, Salomon Eliasson, Erwin Wolters, Ronald Scheirer

**Reviewer 2**

**General comments**

This study aims to simulate AVHRR radiances, with subsequently produced cloud products, from VIIRS radiances with similar satellite orbital configuration. The stated goal is extension of the CLARA CDR, and I believe the potential benefits and uses would extend beyond that. The authors' arguments are well constructed with conscious use of independent validation data, and the writing is strong. The clear presentation of results with added exploration of cloud parameter validation is appreciated.

I think the one major area of improvement falls in the area of the linear SBAF methodology explanation. It is stated that the linear SBAF is determined by looking at simultaneous observations, but it isn't clear to me how the authors would be able to separate spectral bias from radiometric bias with this approach. That is, the inter-calibration sequence has three components to account for, 1) radiometric bias, 2) spectral bias, 3) retrieval biases. The authors mitigate possible retrieval biases with careful angle-matched colocation, and then seem to claim that any remaining bias is explained by spectral differences. If that's true, then the authors would be suggesting that AVHRR and VIIRS have absolute radiometric consistency that is stable across the record. If this assumption is the case, then the authors should make that clear and share their justification.

Regarding discussion of the NASA SBAF tool: It should be noted that the NASA-derived SBAFs are limited by the fact that IASI covers the continuous spectral range of 3.60 - 15.50 µm, and thus computations that consider spectral response below 3.60 µm must rely on assumptions, which are likely imperfect. This should not be framed as a "problem" (Line 146) with the NASA-derived SBAFs, but rather a knowledge limitation due to an observation gap. The AVHRRs and VIIRS have significant response below 3.6 µm, thus computing SBAFs for these would be questionable to begin with. That is, if the use of such would not be recommended, then their impact should not be framed as "negative" (Line 146).

I recommend acceptance after the above minor issues are addressed, as well as the minor specific comments below.

Author response: *We thank the reviewer for these appreciating words. We notice that the main issues to discuss and improve upon aligns well with comments from Reviewer 1, so for parts of our reply we will refer to (or copy) the response to Reviewer 1. We provide our answers and explanations below.*

**Specific Comments:**

Line 87: "satCORPS" should be identified as "satellite cloud and radiation property retrieval system (SatCORPS)"

Author response: *Yes, we have changed this now.*

Line 139-141: Given the limited spectral range of IASI data combined with the effects of solar contribution in the 3B channel range, "serious deviations" should not necessarily be unexpected (see discussion above).

Author response: *Yes, we admit it was a too strong statement. We propose the following instead on line 139 and onwards:*

"This study initially tested various SBAF relations, primarily sourced from NASA (2016). Results were acceptable for most AVHRR channels, but for some channels (especially channel 3B at 3.7 µm), we encountered problems in using the VIIRS-based simulations. For example, night-time cloud detection significantly overestimated the low-level cloud amount. The cloud detection method used (CMAPROB, described by Karlsson et al., 2020) is a probabilistic method using all AVHRR channels. AVHRR channel 3B is considered the most crucial channel for this method's performance, especially at night. Only minor deviations from the original AVHRR channel 3B radiances significantly affect the results at night. The encountered problems were likely caused by the limitation of IASI not observing radiances for wavelengths shorter than 3.62 µm. Since the AVHRR channel 3B and VIIRS band M12 spectral responses both allow for significant contributions at wavelengths shorter than 3.62 µm (see Figure 2), this limitation can be substantial, especially since this affects in particular the contribution from reflected solar radiation, which rapidly increases with decreasing wavelengths. An effort to describe these contributions using Radiative Transfer Model (RTM) calculations was applied in the satCORPS tool, but this was made using assumptions, making results more uncertain."

Line 146: See discussion above regarding "problems" and "negatively."

Author response: *We changed the sentence on line 146 to the following:*

"Due to the uncertainties encountered for the NASA-derived SBAFs for this channel, we decided to proceed by calculating SBAFs from collocated AVHRR- and VIIRS-observed radiances."

Table 1: What's the reason for using Training 1, Training 2, and Validation as a naming convention rather than the more typical Training, Validation, and Testing?

Author response: *We cannot see that there is an obvious naming convention to be used here. For example, the "Testing" option is unclear to us (e.g., what's the difference from "Validation"?). However, we believe that what confuses the Reviewer is the indication that we use two different training datasets. But we think that everything should be clear from the statements made on lines 181-183 (repeated here):*

"The linear SBAF regression methods utilized the entire training dataset (dataset 1 and 2). For the Neural Network (NN) approach, training dataset 1 was used for the actual training and dataset 2 was used as a "during training validation dataset" to decide when to stop the training. The radiance validation dataset was used to evaluate the performance of all SBAF approaches."

*Thus, the confusion probably comes from the fact that Neural Network training was done in two steps (as explained), where the second step involved an evaluation procedure to decide when to stop the training. We believe that the explanations on lines 181-183 should suffice for the understanding of the training, testing, and validation procedures.*

Line 233: Please define "channel quota."

Author response: *It is simply one channel divided by another (e.g. M10/M6 or Ch3a/Ch1). An example will be given in brackets on line 233. However, we will now use the notation "channel ratio" in the manuscript to clarify it even further.*

Table 5: Were other activation functions tested, e.g., ReLU?

Author response: *No, we just kept the activation function we normally use. Different activation functions could be tested in the future, but we do not expect that to make a large difference.*

Lines 278-281: The authors should explain why > 0.20 and > 0.4 were specifically used. Was a sensitivity analysis done?

Author response: *Reviewer 1 had the same question. This is our reply:*

*The COT threshold of 0.2 for validation of CFC is well-documented, e.g., in the CLARA-A2 validation paper by Karlsson and Håkansson from 2018. Notice the improvement from an earlier value of 0.3 as reported by Karlsson and Johansson from 2013 ([https://amt.copernicus.org/articles/6/1271/2013/](https://amt.copernicus.org/articles/6/1271/2013/)) that was seen when going from CLARA-A1 to CLARA-A2. The value has improved further for CLARA-A3 but only marginally. Down to this threshold value, clouds are generally detected well by the current CMAPROB method, i.e., with 50 % or higher probability, and the use of this threshold consequently gives the best and most reliable validation scores for CFC compared to not doing any COT thresholding at all. But, if we are able to detect thinner and thinner clouds, it doesn't automatically also lead to improvements CTH improvements, simply because the complexity for the semi-transparency correction increases for thinner clouds.*

*The threshold to be used for CTH validation is currently more arbitrarily chosen. It should be higher than 0.2 but not drastically higher, since cloud detection efficiency increases rapidly for COTs larger than 0.2 (as illustrated by Karlsson and Håkansson, 2018, Figure 6). A value of 0.4 was found reasonable to at least avoid some of the uncertainties with very thin clouds. At the same time, the threshold should not be too high, so that it does not give justice to all efforts to correct for semi-transparency (i.e., it would not make sense to only look at opaque clouds). We could have produced results for a wide range of COT thresholds, but we found it appropriate to stay with the established method that we have recently used regularly for evaluation of results from CLARA-A2 and CLARA-A3 in standard EUMETSAT validation and review studies. We also don't think that results should be restricted to only include COT-thresholded results, since it always has a value to also see unfiltered results, as it is generally easier to see improvements when introducing new or upgraded methods (at least for CFC).*

*We also introduced the following comment on line 400:*

"The use of different COT thresholds here is motivated by the wish to avoid the thinnest clouds detected by the cloud masking procedure, since these are always the most difficult clouds to deal with for the CTH retrieval."

*As a last remark on this issue, we remind the reviewer that we have made a small change to the content of tables 7-9 after discovering that the category "No SBAF" was not adequately described in the first version of the manuscript. As a consequence, two alternative interpretations of "No SBAF" have now been introduced in the revised tables. For further clarification, the reviewer is asked to read the discussion of tables 7-9 in the reply to Reviewer 1.*

Line 301: Table 2 shows the spectral relationship of VIIRS and AVHRR channels. It does not explicitly show how radiances compare unless such is broadly inferred from the spectral information (which would be scene-dependent). The authors should clarify how the table shows the radiance comparison, or is this a mistake?

Author response: *This sentence is definitely wrong. Thanks for discovering, it is now removed.*

Lines 543-544: This concluding statement regarding "implicit object type identification" is excellent.

Author response: *Thank you very much!*

Lines 545-547: I agree that using more VIIRS channels in the NNet would lead to better-simulated AVHRR channels with improved consideration of specific scene types. Is there a reason the authors did not use additional training channels to being with? There is seemingly no need to limit selection to comparable channels when it comes to training purposes.

Author response: *The reviewer is absolutely right in that theoretically one could have used all available VIIRS channels for the training. But for practical reasons, adding more input will also need more processing time and more processing resources. However, we did test various setups and the chosen one seemed to be the most appropriate setup. The various tests were actually only with fewer channels than what was finally used. We started with the channels we definitely needed, then added the 8.5 μm channel to have one more channel during night where cloud products initially looked worse. Finally, we added the 1.6 μm channel to get one more visual channel for day/twilight. The processing time is maybe not the biggest problem, but with every new channel added there is a need to do validation to make sure we do not get unexpected results and new problems.*

Lines 572-573: It should be mentioned that the basic radiances also would need to maintain stability.

Author response: *This was also one of the major points brought up by Reviewer 1. We are now adding the following sentences in the Conclusions section:*

"
In order to check the validity of the derived spectral adjustments on a longer timescale, it is suggested to regularly repeat the validation efforts on derived cloud products based on new observations from active sensors on EarthCARE (Illingworth et al., 2015) and similar missions in the future. In addition, regular checks of Simultaneous Nadir Observations (SNOs) at high latitudes between VIIRS and IASI (and its successors) can also help in deducing the infrared bands spectral responses' stability. The possibility to make similar checks for VIIRS and METimage visible bands seems unfortunately not possible from Troposphere Ozone Monitoring Instrument (TROPOMI, Veefkind et al., 2012) due to very limited spectral coverage of the AVHRR-heritage channels. However, it is possible that measurements from the Ocean Color Instrument (OCI) sensor on the Plankton, Aerosol, Cloud, and ocean Ecosystem (PACE, Gorman et al., 2019) mission and the Earth surface Mineral dust source Investigation (EMIT, Connely et al., 2021) sensor onboard the International Space Station (ISS) can be used. Some reference measurements from the Sentinel 3 Ocean and Land Colour Instrument (OLCI, Donlon et al., 2012) measurements could also be used. Although not purely hyperspectral, OLCI contains 21 bands in the 0.4-1.0 μm range and has an equator overpass time at 10:00 AM."

---

## Referee Report (RR1)

**Review of egusphere-2025-379 version 2**

**Extension of AVHRR-based climate data records: Exploring ways to simulate AVHRR radiances from Suomi-NPP VIIRS data**

By: Karl-Göran Karlsson, Nina Håkansson, Salomon Eliasson, Erwin Wolters, Ronald Scheirer

**Overall Recommendation**

The authors did an excellent job in revising version 1 of their paper, they clearly replied to all my points of criticism and modified the manuscript accordingly. The revised version improved a lot, both in terms of information content and clarity, as well as in the presentation of the information.

I have found a few minor points of concern, which I mention below. Apart from these points, the manuscript is suitable for publication as far as I am concerned.

**Minor Criticisms**

*Table 7, 8, 9:* The authors give a very extensive reply to my question on the differences seen in validation results of the CFC results for COT>0.0 and COT >0.2 and the CTH results for COT >0.0 and COT>0.4. The basic message of my comment was: Why do the authors keep in the results of the COT>0.0, and not only present the CFC results for COT>0.2 and CTH results for COT>0.4?

I am happy with the reply of the authors, and it is fine to keep both. However, the differences between the results of COT>0.0 and COT>0.2 or 0.4 may raise many questions among readers. Thus, some additional explanation in the text would be helpful, can the authors add the main message of their reply, in a condensed form, as explanatory text to the results of Table 7, 8, 9?

*Table 6:* A % sign is missing

*Table B1:* The updates made to the tables in Appendix B are great, they make the picture complete. I noticed that the text of caption does not yet match with the updated content of the table (eg it mentions RMSE and does not mention CORRELATION). Please check and adjust

---

## Author Response (AR2)

**Review of egusphere-2025-379**
**Extension of AVHRR-based climate data records: Exploring ways to simulate AVHRR radiances from Suomi-NPP VIIRS data**

By: Karl-Göran Karlsson, Nina Håkansson, Salomon Eliasson, Erwin Wolters, Ronald Scheirer

**Comments from Reviewer 1 and the Editor**

Comments from Reviewer 1:

Overall Recommendation

The authors did an excellent job in revising version 1 of their paper, they clearly replied to all my points of criticism and modified the manuscript accordingly. The revised version improved a lot, both in terms of information content and clarity, as well as in the presentation of the information.

Author response: Thank you very much!

I have found a few minor points of concern, which I mention below. Apart from these points, the manuscript is suitable for publication as far as I am concerned.

Minor Criticisms

Table 7, 8, 9: The authors give a very extensive reply to my question on the differences seen in validation results of the CFC results for COT>0.0 and COT >0.2 and the CTH results for COT >0.0 and COT>0.4. The basic message of my comment was: Why do the authors keep in the results of the COT>0.0, and not only present the CFC results for COT>0.2 and CTH results for COT>0.4?

I am happy with the reply of the authors, and it is fine to keep both. However, the differences between the results of COT>0.0 and COT>0.2 or 0.4 may raise many questions among readers. Thus, some additional explanation in the text would be helpful, can the authors add the main message of their reply, in a condensed form, as explanatory text to the results of Table 7, 8, 9?

Author response: *We can certainly do that. We suggest to rewrite the entire paragraph before Table 6 (notice that one table was removed in the updated manuscript, thus validation result tables are now Tables 6, 7 and 8) as follows (starting at line 438):*

"The tables also show the original requirements for the three cloud products in the CLARA-A3 CDR in the rightmost columns. Products generated from VIIRS/VGAC-simulated data should also fulfill these requirements. In the CLARA-A3 evaluation, very thin clouds detected by CALIPSO-CALIOP were removed based on COT thresholds of 0.2 and 0.4 for CFC and CTH, respectively. Karlsson and Håkansson (2018) suggested this CFC COT threshold after studying the effect of COT thresholding during the CLARA-A2 CDR CFC validation exercise. They found the best overall validation scores when excluding clouds with COT lower than this threshold. This COT-thresholding was later used to define the CLARA-A3 CFC requirement (Table 6, rightmost column). Here, using different CFC and COT thresholds (Tables 6 and 7) is motivated by wanting to discard the thinnest clouds when validating the CTH product. These clouds are always the most difficult to deal with for any CTH retrieval, i.e., the thinner the clouds, the more challenging it becomes to compensate for semi-transparency effects. The

COT threshold for CTH validation is more arbitrarily chosen. It should be higher than 0.2, although not drastically, since cloud detection efficiency increases rapidly for COTs larger than 0.2 (see Karlsson and Håkansson, 2018). A reasonable threshold of 0.4 was found to remove some of these thin cloud uncertainties. The threshold should not be too high to give justice to all semi-transparency correction efforts, i.e., it would not make sense to only look at opaque clouds. To highlight improvements more clearly, Tables 6 and 7 show both the COT-thresholded results, used for CLARA-A3 requirements, and the original results that include optically thin clouds. Relying only on the COT-thresholded results would overlook important improvements to the overall CDR, particularly for the CFC product."

*Just to comment a bit more what this COT-thresholding brings to the results: For CFC it really means that most of the differences caused by the higher sensitivity to thin clouds for CALIOP disappear (i.e., the bias is reduced to almost zero %, thus well within the requirements of 5 %). For CTH, the effect is less obvious (i.e., requirements are hardly met) but this is mostly explained by the problems of using the mean error score as the basis for the requirement. It should rather have been the mean absolute error. This is clearly mentioned in the text so no further changes are needed.*

Table 6: A % sign is missing

Author response: *Corrected (Column 4, Row 5 in Table 6).*

Table B1: The updates made to the tables in Appendix B are great, they make the picture complete. I noticed that the text of caption does not yet match with the updated content of the table (eg it mentions RMSE and does not mention CORRELATION). Please check and adjust"

Author response: *Indeed, we missed this aspect. Now corrected in the captions to Tables B1-B4.*

Comments from Editor:

"Checking your paper, I noticed that your Table 2 contains coloured text. Please note that this will not be possible in the final revised version of the paper due to HTML conversion of the paper. When revising the final version, you can use footnotes or italic/bold font."

Author reply: *This is now corrected. We have used bold fonts as replacement for the colour.*

Additional changes:

Author response: *There are a few typos corrected and some small editorial changes made in the manuscript (visible in the tracked changes version of the document).*